https://doi.org/10.1038/s41467-021-21079-6　　**OPEN**

# C-STABILITY an innovative modeling framework to leverage the continuous representation of organic matter

Julien Sainte-Marie [1,2✉], Matthieu Barrandon[3], Laurent Saint-André[2], Eric Gelhaye[4], Francis Martin[4,5] & Delphine Derrien[2✉]

The understanding of soil organic matter (SOM) dynamics has considerably advanced in recent years. It was previously assumed that most SOM consisted of recalcitrant compounds, whereas the emerging view considers SOM as a range of polymers continuously processed into smaller molecules by decomposer enzymes. Mainstreaming this new paradigm in current models is challenging because of their ill-adapted framework. We propose the C-STABILITY model to resolve this issue. Its innovative framework combines compartmental and continuous modeling approaches to accurately reproduce SOM cycling processes. C-STABILITY emphasizes the influence of substrate accessibility on SOM turnover and makes enzymatic and microbial biotransformations of substrate explicit. Theoretical simulations provide new insights on how depolymerization and decomposers ecology impact organic matter chemistry and amount during decomposition and at steady state. The flexible mathematical structure of C-STABILITY offers a promising foundation for exploring new mechanistic hypotheses and supporting the design of future experiments.

[1] Université de Lorraine, AgroParisTech, INRAE, SILVA, F-54000 Nancy, France. [2] INRAE, BEF, F-54000 Nancy, France. [3] Université de Lorraine, CNRS, IECL, F-54000 Nancy, France. [4] Université de Lorraine, INRAE, IAM, F-54000 Nancy, France. [5] Beijing Advanced Innovation Center for Tree Breeding by Molecular Design, Beijing Forestry University, Beijing, China. ✉email: julien.sainte-marie@agroparistech.fr; delphine.derrien@inrae.fr

Soil organic matter (SOM) is the largest reservoir of organic carbon (C) on land[1]. Understanding and modeling the processes driving its dynamics is essential to predict SOM response to changes in climatic conditions and human land management, and its contribution to soil functions and climate mitigation[2].

Our view of the nature of SOM and decomposition pathways has recently been challenged by the wealth of information generated by new analytical techniques[3]. It was previously assumed that most SOM consisted of recalcitrant litter material and humic molecules formed by the condensation of decaying substrates. The emerging view is now that SOM occurs as a range of organic compounds continuously processed into smaller molecules by microorganisms through the production of extracellular enzymes. Under this new SOM cycling paradigm the local environment is seen as a critical driver of SOM decomposition as it determines its physicochemical accessibility to extracellular enzymes and modulates microbial metabolism[4–7]. Metabolic activity of microbes consists of catabolism, in which extracellular enzymes depolymerize SOM, and also of anabolism, which is responsible for $CO_2$ emissions and biochemical transformation of assimilated compounds. The biosynthesized microbial metabolites are known to strongly interact with protective mineral surfaces and may be sequestered for a long time in the soil[6,8–10]. In the last decade new information on the functional diversity of microbial communities and on their catabolic action has become available. Genome analyses notably enable characterization of the enzyme sets encoded by each decomposer species and classification of decomposer taxa into functional communities such as guilds[11–15]. Enzymes production patterns can also be monitored over time via proteome and secretome analyses[16,17].

How have models of SOM dynamics evolved in recent years to incorporate novel knowledge and theories? A number of new compartment models better representing microbial ecology processes have been proposed[18,19]. The most often incorporated features include decomposer physiological factors (carbon use efficiency, growth and mortality rates, etc.) and extracellular enzyme properties (production rate, catalytic properties, etc.)[20–22]. The microbial response to fluctuating environmental parameters such as temperature and nitrogen availability is also sometimes integrated[23]. Some models explicitly describe how the soil porosity and micro-architecture govern decomposer access to substrates[22,24,25] or specify the SOM protection mechanism[26,27]. Yet these compartment models are limited to report the recent insights gained in the fields of functional ecology and biogeochemistry because they assume that SOM can be represented as a few discrete pools with differing turnover times[3]. Besides, increasing the description of compounds and processes in compartment models raises the number of required parameters and leads to model prediction uncertainty[28]. In contrast to compartment models, continuous models have received little attention from developers in recent years. Existing models represent SOM as a distribution along a quality axis and describe with a few parameters how organic matter quality evolves toward more recalcitrant—or stable—stages as decomposition progresses. This is done without a detailed description of microbial and biogeochemical mechanisms[29–32]. At first glance, continuous models seem particularly appealing for representing the current view of SOM as a range of heterogeneous organic compounds at different decomposition stages with few parameters. But they have three major shortcomings. First, quality is viewed as an intrinsic property of organic matter, generally in line with the outdated recalcitrance notion. Second, quality is a concept defined relative to a uniform decomposer group at steady state in existing models, which is unrealistic. These two characteristics hinder the investigation of issues related to functional ecology and decomposer community succession. Third, the notion of quality is complex, not measurable and this abstraction is a major barrier to wide acceptance of the continuous modeling approach.

Here we propose a new general model of SOM dynamics combining advantages and suppressing drawbacks of the current compartment and continuous models. This Carbon Substrate Targeted AccessiBILITY (C-STABILITY) model is driven by microbial activity and combines compartmental and continuous approaches. The substrate is split into biochemical classes, separating pools accessible to enzymes from inaccessible ones (Fig. 1a). Within each pool a continuous approach is implemented to describe the level of substrate polymerization (Fig. 1b). The model distinguishes between substrate accessibility to microbe uptake, only possible for small oligomers, and substrate accessibility to enzymes regulated by organic matter spatial arrangement within the soil matrix or interactions with other soil components[33,34]. Over time the enzyme-accessible substrate is fragmented and depolymerized until it eventually becomes accessible to microbe uptake. Part of the taken up substrate is assimilated by decomposers and its biochemistry is modified by decomposer anabolism (Fig. 1a). One of the major features of the model is the description of the enzymatic polymer breakage process. A substrate cleavage factor, $\alpha_{enz}$, is introduced for each enzyme family (e.g., cellulolytic, proteolytic, etc.) to indicate if it depolymerizes the substrate by cleaving its end-members or if it randomly breaks any bond (Fig. 2). This general model enables a parsimonious representation of the SOM continuous nature and degradation pathways by functional decomposer communities as they are currently understood (Supplementary Table 1).

In this article, we explore C-STABILITY predictions regarding the following key-questions. (1) Is enzyme depolymerization a critical regulator of SOM decomposition? (2) How is substrate accessibility to enzyme regulated? (3) How does the succession of decomposer communities impact the chemistry of the decaying substrate? (4) Is microbial biomass the dominant source of the SOM stock? We address these questions by analyzing four realistic scenarios designed on the basis of recent publications (see Methods and Supplementary Table 1). Two scenarios focus on substrate decomposition: the first involved a homogeneous simple substrate, the second a more complex substrate. The two last case studies focus on microbial recycling, simulating either substrate decomposition kinetics or steady state. Finally, we discuss the opportunities offered by the innovative framework of C-STABILITY to improve our understanding of organic matter cycling in soil.

## Results

**Depolymerization pattern as a major driver of organic matter accessibility to uptake by microbe**. We ran a first scenario of cellulose decomposition to investigate if enzyme depolymerization, reported in an original manner by the $\alpha_{enz}$ parameter in C-STABILITY, is a crucial regulator of substrate decomposition kinetics compared to other well-known drivers such as carbon use efficiency or enzyme action rate (also called catalytic rate). A cellulose polymer was subjected to the cellulolytic action of a set of enzymes and split into smaller fragments, which corresponded to a gradual decrease in the cellulose polymer size distribution toward short oligomers in the microbial uptake domain, $\mathcal{D}_u$ (Fig. 3a). Cellulose oligomers small enough to be assimilated were continuously produced and almost fully taken up with the chosen parameters (Table 1). They were used either to produce microbial biomass or to generate energy trough the respiration process (Fig. 3b). Ninety-five percent of the cellulose-C was consumed after 123 days, with a mean residence time of 97 days, which was consistent with results of laboratory incubation studies[35].

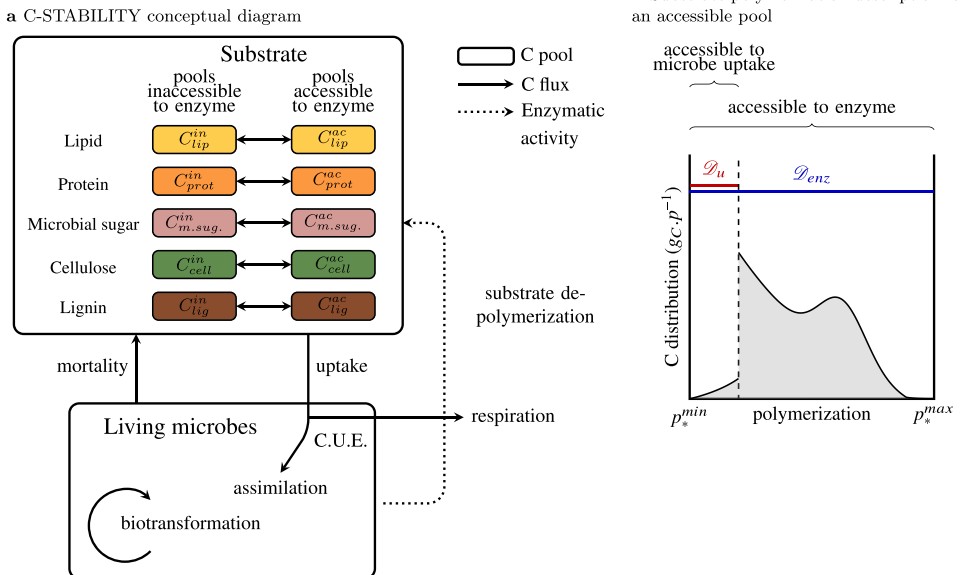

**Fig. 1 C-STABILITY model framework. a** Conceptual diagram presenting interactions and exchanges between living microbes and organic substrate. Colored boxes stand for substrate C pools. The colors represent the different biochemical classes. The forms accessible to enzymes are separated from the inaccessible ones. Plain arrows represent C fluxes, the dotted arrow represents enzymatic activity. Microbial uptake is only possible for small oligomers, which are produced by enzymes. C.U.E. means carbon use efficiency. Assimilated C is biotransformed by microbes and returns to substrate through mortality. **b** Substrate polymerization (noted $p$) is described for each C pool. Polymerization is used to define how living microbes get access to substrate uptake. A continuous distribution reports the polymerization level of the C substrate of each biochemical class (denoted *). $p_*^{min}$ and $p_*^{max}$ are respectively the minimum and maximum levels of polymerization. The polymerization axis is oriented from the lowest polymerization level on the left to the highest polymerization level on the right. Here an accessible pool is presented. The same definition of polymerization stands for pool accessible and inaccessible to enzymes. The blue domain $\mathcal{D}_{enz}$ identifies polymers accessible to enzymes. The red domain $\mathcal{D}_u$ identifies monomers and small oligomers accessible to microbe uptake. The amount of C corresponds to the area below the curve.

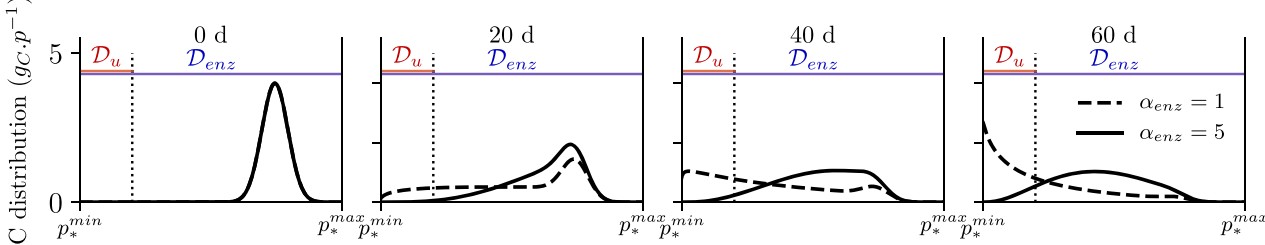

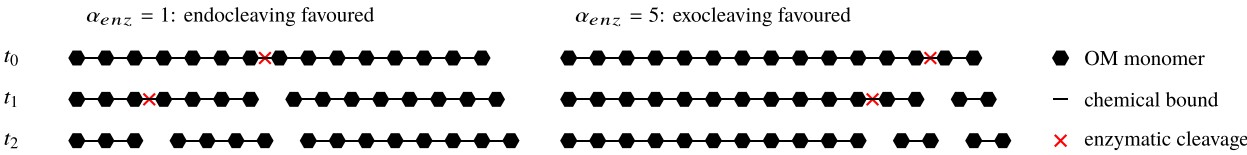

**Fig. 2 Change in substrate polymerization over time induced by enzymes for an accessible biochemical class (denoted *) in the absence of microbial uptake. a** A continuous distribution reports substrate polymerization, noted $p$. $p_*^{min}$ and $p_*^{max}$ are the minimum and maximum levels of polymerization. The $\mathcal{D}_{enz}$ domain (blue) indicates substrate accessible to its enzymes. The $\mathcal{D}_u$ domain (red) indicates where the substrate is accessible to microbe uptake. Here the microbe uptake rate is nil, which explains the substrate accumulation in $\mathcal{D}_u$. **b** Two substrate cleavage factor $\alpha_{enz}$ values are tested. $\alpha_{enz} = 1$ is typical of the action of endo-cleaving enzymes, which randomly disrupt any substrate bond and generate oligomers, inducing a rapid shift toward the $\mathcal{D}_u$ microbial uptake domain. $\alpha_{enz} = 5$ is typical of exo-cleaving enzymes attacking the substrate end-members to release small oligomers, inducing a slower shift toward $\mathcal{D}_u$.

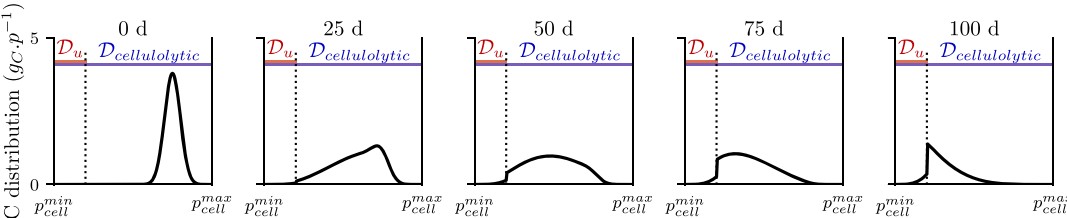

**a** Changes in cellulose polymerization distribution over time

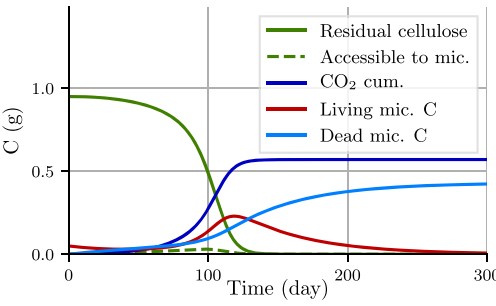

**b** Changes in C stocks during cellulose degradation

**c** Parameters affecting the amount of residual cellulose

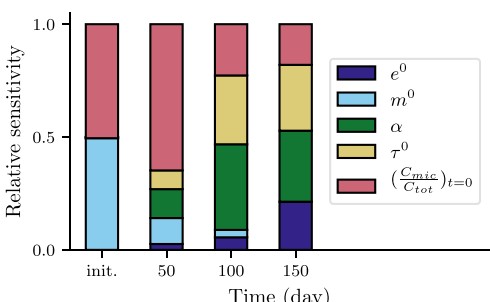

**Fig. 3 Dynamics of cellulose degradation by one decomposer community (scenario 1). a** Changes in the distribution of cellulose polymer length over time, reported in $g_C.p^{-1}$ with $p$ standing for the substrate polymerization. Cellulose is progressively depolymerized by enzymes. The $\mathcal{D}_{cellulolytic}$ domain (blue) is indicative of cellulose accessible to cellulolytic enzymes, the $\mathcal{D}_u$ domain (red) is indicative of cellulose oligomers accessible to microbe uptake; $p_{cell}^{min}$ and $p_{cell}^{max}$ are the minimum and maximum levels of cellulose polymerization. The distribution shift toward $\mathcal{D}_u$ over time illustrates the depolymerization of cellulose. Microbial uptake induces a substrate depletion in $\mathcal{D}_u$ and causes a distribution discontinuity. **b** Amount of residual cellulose-C, C available for microbial uptake, cumulated $CO_2$, living biomass-C, and dead microbial residues-C. There is no microbial recycling in this simulation. Note that the sum of cellulose-C and living microbes-C at the beginning is equal to the sum of cumulated $CO_2$ and microbial residues-C at the end. **c** Sensitivity analysis on the amount of residual cellulose-C (±5% uniform variability of the parameters) at different times. Relative importance of model parameters changes over time depending on their role in the substrate degradation process. Tested parameters are the carbon use efficiency $e_{mic}^0$, mortality rate $m_{mic}^0$, substrate cleavage factor $\alpha_{enz}$, enzyme activity rate $\tau_O$, and initial microbial-C to total-C ratio. For instance $\alpha_{enz}$ has no influence at the beginning while it explains more than 40% of the total model variance at 100 days.

Note that during the first 25 days the enzyme action mostly generated oligomers that were too large to be accessible to microbe uptake, inducing a decrease in the living microbe biomass (Fig. 3a, b).

The sensitivity analysis of parameters shows that their relative influence on the amount of residual cellulose varied over time (Fig. 3c). As the decomposition dynamics was driven by microbes, any modification affecting the size of their biomass had a marked impact on the residual cellulose amount. Thus, the mortality rate $m_{mic}^0$ and the initial $C_{mic}/C_{tot}$ ratio had a strong influence at the beginning of the simulation, while the microbial carbon use efficiency $e_{mic}^0$ was a sensitive parameter at the end of the simulation. The enzymatic substrate cleavage factor $\alpha_{enz}$ and enzyme action rate $\tau_{enz}^0$, which controlled the amount of substrate accessible to microbe uptake, became crucial parameters once the microbe community started growing. The decrease in the $\alpha_{enz}$ value induced a faster depolymerization pattern and made the substrate accessible to decomposer uptake more rapidly (Fig. 2). $\alpha_{enz}$ indeed indicated whether the enzyme preferentially attacked substrate end-members (exo-cleaving enzyme – high $\alpha_{enz}$ values) or randomly disrupted any substrate linkage (endo-cleaving enzyme – low $\alpha_{enz}$ values). Exo-cleaving enzymes released monomers or dimers, which caused a gradual decrease in the polymer length (Fig. 2). Endo-cleaving enzymes efficiently shifted the polymer size distribution toward short lengths. Variations in the microbe uptake rate $u_{mic}^0$ had a negligible impact on the model sensitivity. Due to the other parameters values, the available C in

$\mathcal{D}_u$ is almost instantaneously taken up, whatever the $u_{mic}^0$ value within a range of ±5% variability around the default value.

This simulation illustrates that C-STABILITY differentiates substrate transformation by enzyme and substrate uptake by decomposers, contrary to other models, which do not represent substrate polymerization and consider implicitly that depolymerization and uptake are simultaneous. This innovation enables us to demonstrate that depolymerization, whose action is reported by the enzyme cleavage factor, is a major limiting step controlling microbe uptake and decomposition of SOM. It has been largely overlooked until now and would deserve more attention in future experimental and modeling works. In addition, reporting the depolymerization process in the model frame permits to describe how the decaying substrate is transformed over time through a distribution of polymer sizes.

**Coordinated action of enzymes regulates substrate accessibility.** We ran a simulation of lignocellulose decomposition to examine the regulation of substrate accessibility to enzyme. In this study case, the activity of one enzyme family provides a gateway to the substrate for another one. Indeed in wood the spatial arrangement and the interactions of cellulose with lignin makes it inaccessible to cellulases. Cellulose depolymerization and utilization can only start once the lignin barrier has been altered. This typically occurs during wood attack by wood-rotting fungi producing lignolytic factors and cellulases, sometimes in a temporal sequence[16,17,36]. In this second scenario, lignocellulose was

**Table 1 Synthesis of C-STABILITY parameters and scenarios setups with values for initial conditions, microbes traits, and enzymes families. Scenario 1 corresponds to cellulose degradation, scenario 2 to lignocellulose degradation, scenario 3 to succession of two different microbial communities on lignocellulose, considering microbial recycling, and scenario 4 to soil organic matter at steady state considering a continuous plant input and microbial recycling, with only one microbial community. Initial amounts of microbes and substrate are respectively noted $(C_{mic})_{t=0}$ and $(C_{sub})_{t=0}$, mic. stands for microbe and sub. for substrate. For scenario 3, two $u^0_{mic}$ values were tested to compare non-cheating and cheating behavior of decomposers. For scenario 4, the values of $\tau^0_{enz}$ were divided by 5 to account for in situ conditions, while other scenarios are based on laboratory case studies (Supplementary Table 1).**

| Parameter | Description | Unit |
|---|---|---|
| $u^0_{mic}$ | Microbial uptake rate per microbial carbon mass | $g_C^{-1}.d^{-1}$ |
| $e^0_{mic}$ | Microbial carbon use efficiency | — |
| $m^0_{mic}$ | Microbial mortality rate | $g_C^{-1}.d^{-1}$ |
| $\tau^0_{enz}$ | Enzymatic action rate per microbial carbon mass | $g_C^{-1}.d^{-1}$ |
| $\alpha_{enz}$ | Enzymatic substrate cleavage factor | — |

**Initial setup for each scenario**

| Scenario | Microbes | Initial conditions ($g_C$) | | Microbial signature | | | Plant biochemistry | |
|---|---|---|---|---|---|---|---|---|
| | | $(C_{mic})_{t=0}$ | $(C_{sub})_{t=0}$ | Lipid | Protein | Mic. sugar | Plant sugar | Lignin |
| 1 | Plant decomposer | 5 | 95 | – | – | – | 100% | – |
| 2 | Plant decomposer | 5 | 125 | – | – | – | 76% | 24% |
| 3 | Plant/mic. decomposer | 5/5 | 125 | 30% | 20% | 50% | 76% | 24% |
| 4 | General decomposer | – | – | 30% | 20% | 50% | 76% | 24% |

**Microbes parameters for each scenario**

| Scenario | Microbes | $e^0_{mic}$ | $m^0_{mic}$ | $u^0_{mic}$ | | | | |
|---|---|---|---|---|---|---|---|---|
| | | | | Lipid | Protein | Mic. sugar | Plant sugar | Lignin |
| 1 | Plant decomposer | 0.4 | 0.02 | — | — | — | 5 | — |
| 2 | Plant decomposer | 0.4 | 0.02 | — | — | — | 5 | 5 |
| 3 | Plant decomposer | 0.4 | 0.02 | 0/3 | 0/3 | 0/3 | 5 | 5 |
| 3 | Mic. decomposer | 0.5 | 0.01 | 5 | 5 | 5 | 0/3 | 0/3 |
| 4 | General decomposer | 0.4 | 0.02 | 5 | 5 | 5 | 5 | 5 |

**Enzymes parameters for each scenario**

| Scenario | Enzymes action | $\tau^0_{enz}$ | $\alpha_{enz}$ |
|---|---|---|---|
| 1-2-3/4 | Cellulolysis | 1.8/0.36 | 5 |
| 2-3/4 | Lignolysis | 1/0.2 | 7 |
| 3/4 | Lipidolysis | 0.8/0.16 | 1.5 |
| 3/4 | Proteolysis | 1.8/0.36 | 5.5 |
| 3/4 | Microbial sugar lysis | 1.8/0.36 | 5 |

made of 76% cellulose, 24% lignin. A specific design of the initial cellulose distribution was made to represent its embedment with lignin, which made it inaccessible to cellulolytic enzymes (Fig. 4a and Supplementary Movie). Parameters were chosen to model a decomposition pattern, in general agreement with the findings of lignocellulose decomposition studies performed in microcosms[37].

Lignin deconstruction was initiated from the beginning of the simulation but the amount of lignin-C did not decrease prior to 120 days (Fig. 4 and Supplementary Movie). Lignin depolymerization was very slow because of the combined effects of the low lignolytic rate $\tau^0_{enz}$ and a high substrate cleavage factor $\alpha_{enz}$ (Table 1). Nevertheless the simulation revealed that the alteration of the lignin framework rapidly opened the wood cell wall structure[38] and cellulose, which was initially inaccessible to cellulolytic enzymes, became completely accessible after 25 days (Fig. 4a). This required step for lignolysis delayed cellulose decomposition compared to the previous simulation (Fig. 3) with a pure cellulose substrate (95% of the cellulose-C is consumed after 140 days versus 123 days in the previous case) (Fig. 4 and Supplementary Movie). Formally, mechanisms regulating cellulose accessibility to cellulases were modeled in C-STABILITY

with a translation of the cellulose polymerization distribution from a pool where it was not accessible to another pool where it became accessible to its enzymes. This was assumed to be a linear function of the lignolytic enzymes activity. The substrate remained unaltered as long as it was inaccessible to enzymes, as also implemented in[39]. This approach conceptually differs from that adopted in many compartment models where inaccessibility is reported by diminishing the substrate turnover rate compared to its free form[20,26,40,41].

This simulation illustrates how the action of one enzymes family on a complex substrate may be a prerequisite to the action of another enzymes family. It demonstrates the capacity of C-STABILITY to report the joint action of specific enzymes and to provide quantitative information about it, which may be hard to capture experimentally. This scenario could be further utilized to theoretically explore different depolymerization strategies implemented by decomposers to degrade complex substrate. For example, tuning the $\alpha_{enz}$ parameter for lignolytic activity would enable to compare brown rot and white rot fungi degradative strategies and evaluate their impact on fungal growth and wood degradation kinetics. Random bond disruption mediated by

**a** Changes in polymerization distribution in cellulose and lignin pools

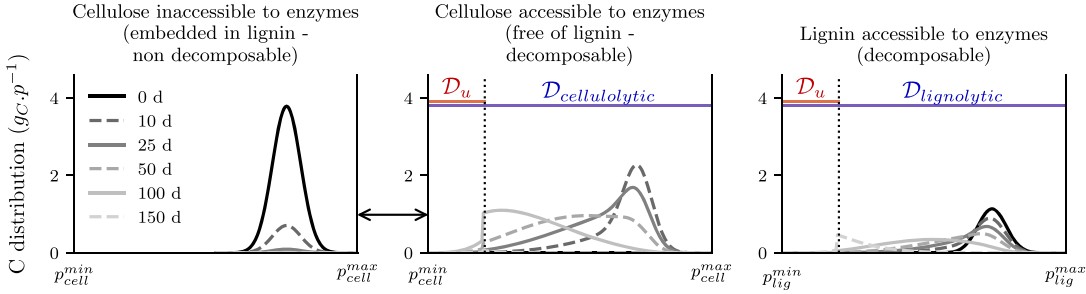

**b** Variations in C stocks over time in cellulose and lignin pools

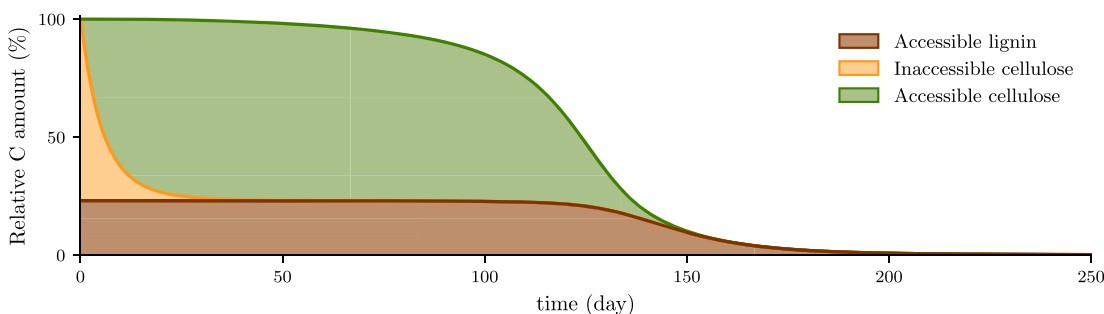

**Fig. 4 Degradation of lignocellulose by a microbe community producing cellulolytic and lignolytic enzymes (scenario 2).** Cellulose is initially embedded in lignin and not accessible to its enzymes. **a** Distribution of polymerization, noted *p*, in the pools of inaccessible cellulose (left), cellulose accessible to cellulolytic enzymes (center), lignin accessible to lignolytic enzymes (right). Domains of substrate accessibility to enzymes are in blue, domains of substrate accessibility to microbial uptake are in red. The black arrow between inaccessible and accessible cellulose highlights the transfer of C between the two cellulose pools. Inaccessible cellulose is progressively transferred in the accessible pool without alteration of its polymerization. The transfer is directly related to lignin depolymerization and the associated degradation of the physical barrier. **b** Residual amount of C in the different pools over time. The lignolytic activity induces a quick disentanglement of the cellulose from the lignocellulosic complex (from orange to green), which makes polymerized cellulose accessible to its enzymes. Depolymerized cellulose and lignin are taken up by microbes. A Supplementary Movie illustrates this scenario.

hydroxyl radical produced by brow rot fungi would be reported by a lower $\alpha_{enz}$ value than enzyme-mediated disruption of bonds by white rot fungi.

**Interactions between successive decomposer communities regulate the persistence of decomposing substrate and its chemistry.** The C-STABILITY framework is tailored specifically to reflect the succession of microbial players during litter decomposition and their biogeochemical functions. In a third scenario we implemented the model with two microbial functional communities succeeding each other on a lignocellulose substrate. We examined how microbes dynamics is regulated by the chemistry of the decaying substrate and conversely how residual substrate chemistry is affected by the interaction between the microbial communities. The two microbial communities had the same biochemistry, consisting of lipids, proteins, and sugars (Table 1), but differed in their catabolic abilities. Recent studies showed that saprophytic fungi are the first to take action in wood decomposition. They are the main producers of enzymes involved in lignocellulose degradation. Bacteria or mycorrhizal fungi peak thereafter, breaking down and utilizing dead fungal biomass rather than plant polymers[42–45]. We referred to them as plant decomposers and microbial residue decomposers, respectively. In our simulation, the two communities also differed in their fitness. Microbial residue decomposers were more competitive than plant decomposers because of their higher carbon use efficiency and

lower mortality rate (Table 1). This succession is illustrated in Fig. 5a. Plant decomposers dominated the microbial biomass during the first 260 days. They fed on the plant small polymers released through the action of the cellulolytic and lignolytic enzymes they produced, until their exhaustion. The microbial residue decomposer community slightly declined during this first stage because of the lack of resources—it had no access to the small plant oligomers. It started growing once the microbial residues were released upon the death of plant decomposers, after around 200 days (Supplementary Fig. 1). At the end of the simulation, there was almost no plant or microbial substrate left.

Despite the strategies implemented by decomposers to achieve privileged access to the substrate they depolymerize, e.g., antibiotics or secondary metabolites secretion[46], bacteria can also proliferate on simple carbohydrates released by fungal enzymes[43]. This has been referred to as cheating behavior. We simulated this behavior by changing the substrate uptake rate of the community that was not producing enzymes from zero to a positive value (Table 1). This change had direct impacts on decomposer communities dynamics and substrate residue chemistry at the end of the simulation. Indeed, when microbial residue decomposers had an opportunity to opt for cheating behavior and benefit from the plant compounds fragmented by plant decomposers, they developed faster than the plant decomposers and rapidly dominated them (Fig. 5b and Supplementary Fig. 1). The decline in fungal communities involved in plant depolymerization downregulated cellulose and

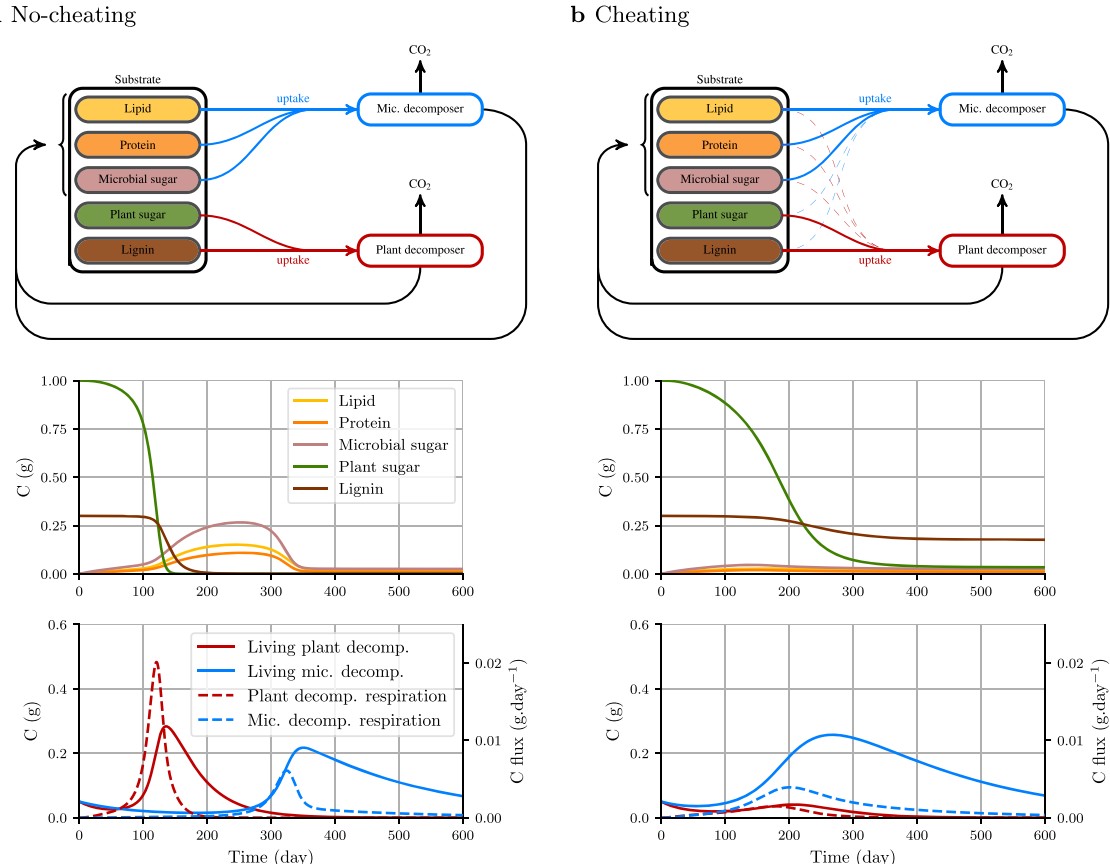

**Fig. 5 Effect of the succession of two decomposer communities on the residual substrate amount and biochemistry (scenario 3).** The two communities have the same biochemical signature (50% sugar, 30% lipid, and 20% protein). Plant decomposers (red) produce cellulolytic and lignolytic enzymes. Microbial residues decomposers (Mic. decomposer = blue) produce enzymes that depolymerize lipids, proteins and microbial sugars. The two communities differ in their fitness. Microbial residue decomposers are more competitive than plant decomposers because of their higher carbon use efficiency and lower mortality rate (Table 1). The effect of cheating behavior was tested as follows: **a** substrate uptake is only possible for the enzyme producer or **b** uptake is also possible for the cheater but at a lower rate than that of the enzyme producer (dashed uptake arrows). Cheating has a strong effect on communities dynamics and the resulting biochemical composition of substrate. When cheating occurs, plant decomposers are outcompeted by microbes decomposers, what results in the persistence of lignin.

lignin depolymerization and disappearance rates. Lignin-C therefore persisted at the end of the simulation (Fig. 5b).

This set of simulations provides novel insights on the processes governing SOM cycling and persistence. C-STABILITY elucidates the dependence of decomposers succession on the availability of accessible substrate, generates the kinetics of decomposer communities succession, and highlights their drivers. It also reveals that a substrate may persist in soil as a partly depolymerized form due to decomposers competition.

**Contributions of plant and microbe residues to SOM at steady state.** Prediction of the contribution of microbial materials to the total carbon stock is of great interest for assessing the C sequestration potential of a system. Indeed microbial residues are currently considered as being the main precursors of stable SOM[8]. For this purpose, we resolved the analytic formulation of the C stock and chemistry at steady state (Eqs. (25), (27), and (28)) while considering one microbial community, microbial residue recycling, continuous plant input and the substrate accessibility to enzymes (scenario 4).

A steady-state C stock of 1.553 $g_C.cm^{-2}$ was computed with the parameters given in Table 1. The living microbial biomass accounted for 0.6% of the total carbon stock (Eq. (25)), which was at the lower limit but within the range observed in soil

across biomes[47]. Cellulose and lignin were the most abundant biochemical substances, each representing 33% of the C stock at steady state (Fig. 6a). Microbial residues contributed to the remaining 33% as follows: 16% microbial sugar, 10% lipids, and 7% proteins. Within each biochemical class, the substrate was a continuum of forms at different decomposition stages. In Fig. 6a, the short peak on the right of the polymerization distribution of SOM corresponded to large polymers initially entering the system as plant litter or microbial residues. Organic matter altered by enzyme action was the main contributor of steady-state SOM, as shown by the dominance of polymers with a lower degree of polymerization. There was no or almost no substrate in the $\mathcal{D}_u$ domains due to active microbial uptake of the smallest substrate fragments. The simulation closely reflected the SOM chemistry observed in situ in boreal soils[48,49]. To predict the SOM chemistry in mineral soil horizons, C-STABILITY should account for the reduction in substrate accessibility due to interactions with the mineral phases, particularly for microbial products that are frequently involved in aggregates or bind on mineral surfaces[8,50–53].

The sensitivity analysis we carried out for the default parameters (Fig. 6b), in a system where substrates were accessible to their enzymes, confirmed the importance of microbial processes in controlling the amount and chemistry of SOM at

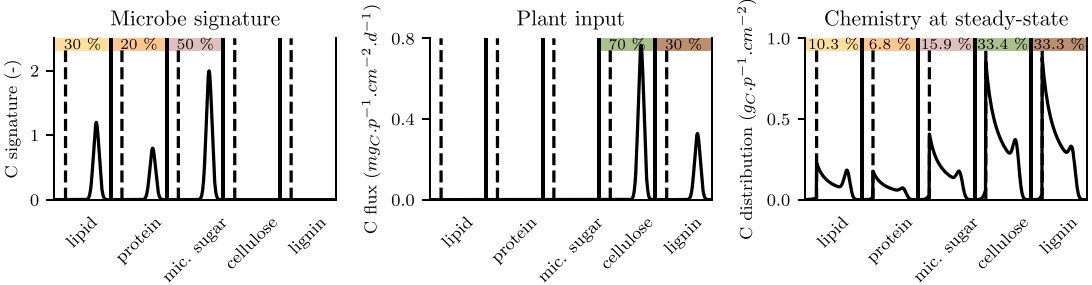

**a** Representation of microbe necromass signature, plant input and SOM at steady state

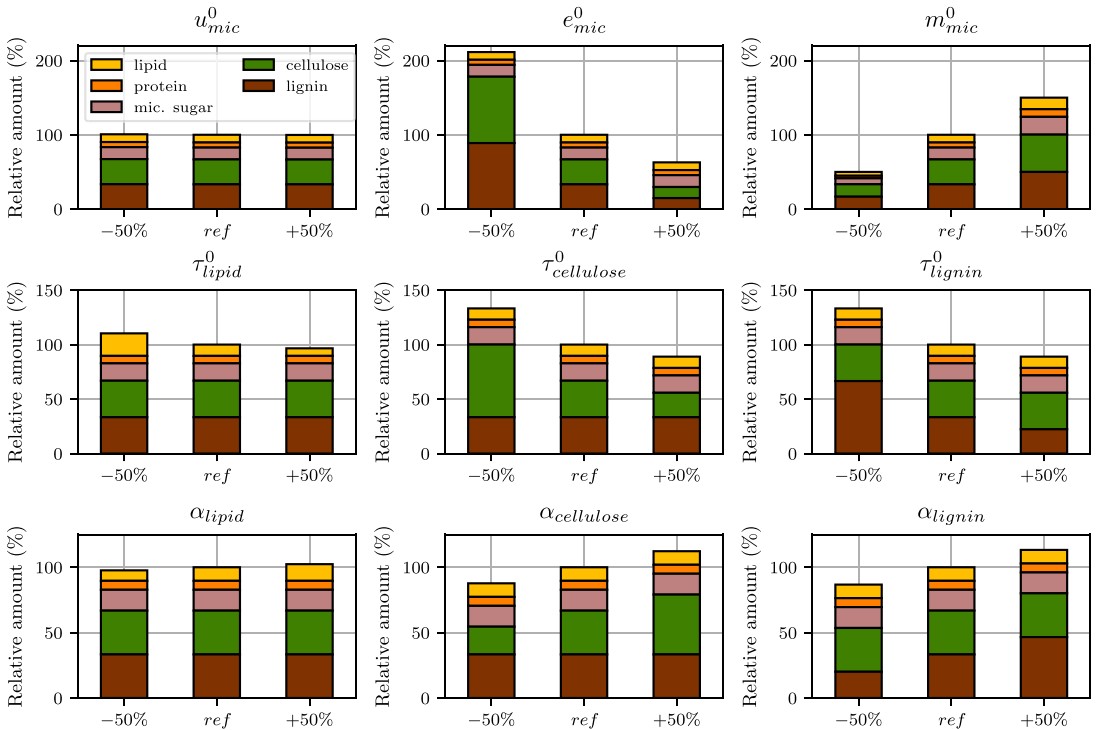

**b** Sensitivity analysis on the C stock at steady state, being 1.553 g$_C$.cm$^{-2}$ with the reference values

**Fig. 6 Soil organic matter at steady state (scenario 4). a** Biochemistry and level of polymerization (noted $p$) of microbe necromass signature (left), plant material input flux (center), and the resulting soil organic matter at steady state (right) with reference parameter values given in Table 1. Different biochemical substrates are separated by vertical solid lines. They are all accessible to enzymes. Microbial uptake domains are positioned at the left of biochemical classes and are delimited by vertical dashed lines. This simulation leading to a C stock of 1.553 g$_C$.cm$^{-2}$ at steady state is considered as reference in the sensitivity analysis. **b** Sensitivity analysis showing the relative changes in total C stock and its distribution among biochemical classes with ±50% changes in the parameters (variations in uptake rate $u_{mic}^0$, carbon use efficiency $e_{mic}^0$, and mortality rate $m_{mic}^0$ at row 1; variations in enzyme action rate $\tau_{enz}^0$ and cleavage factor $\alpha_{enz}$ illustrated for lipid depolymerases, cellulolytic and lignolytic enzymes at rows 2 and 3, respectively). For instance, a reduction of 50% of the value of $m_{mic}^0$ leads to a decrease of 50% of the steady-state C stock in comparison to reference simulation. In this case, the biochemical relative composition remains stable. Note that the scale of the y-axis varies.

steady state. As widely reported[9], the carbon use efficiency parameter, $e_{mic}^0$, which was set to the same value for all biochemical classes (Table 1), was a major driver of C amount. C stock varied in the opposite direction to that of $e_{mic}^0$: total C declined by 37% with a 50% raise in $e_{mic}^0$, while it increased by 111% with a 50% decrease in $e_{mic}^0$. The increase in $e_{mic}^0$ raised the microbial biomass at the expense of $CO_2$ production, which accelerated the C turnover rate through increased substrate depolymerization and uptake (Eqs. (5) and (18)). The steady-state amount of C resulting from the increased value of $e_{mic}^0$ was lower and exhibited a larger relative contribution of microbe-derived metabolites. When looking at the individual biochemical classes

sensitivity to $e_{mic}^0$ we noticed that the microbe-derived compounds were not impacted by variation in $e_{mic}^0$. The accelerated SOM cycling induced by raising $e_{mic}^0$ was indeed exactly counterbalanced by a higher rate of microbial necromass input into the SOM pool with the mathematical formulation and parameters currently implemented in C-STABILITY (Eqs. (27) and (28)). The C stock was also sensitive to the mortality rate $m_{mic}^0$ and exhibited a strong positive linear dependence on it, without any distinction among the various substrate biochemistries. In contrast, the C stock and biochemistry were quite insensitive to changes in the $u_{mic}^0$ uptake rate for the range of tested values (Fig. 6): C uptake was indeed limited here by the

amount of accessible C. Consequently, the enzyme parameters controlling the amount of accessible substrate for microbe uptake had a marked impact on the C stock and biochemistry (Fig. 6b). An increase in the action rate of any enzyme, $\tau_{enz}^0$, decreased the quantity of its substrate at steady state, and vice versa. All biochemical substrates exhibited the same sensitivity to their own enzyme action rate (Eqs. (27) and (28)). An increase of 50% in any given $\tau_{enz}^0$ caused a 33% decrease in the C amount of the biochemical class impacted by the enzyme, while a 50% decrease doubled the C amount of the considered biochemical class. A variation in the cleavage factor of any enzyme, $\alpha_{enz}$, caused a linear variation in the amount of its substrate at the steady state. In term of mechanisms, an increase in $\alpha_{enz}$ corresponded to an enhanced substrate exo-cleavage, which promoted the release of small fragments and preserved large polymers. The sensitivity to $\alpha_{enz}$ differed amongst biochemical classes (Eqs. (27) and (28)). For example a 50% variation in $\alpha_{cellulose}$ caused a 37% in the cellulose-C amount, and a 50% variation in $\alpha_{lipid}$ caused a 22% in the lipid-C amount.

The sensitivity analyses realized on C-STABILITY steady-state simulations question our current perception of SOM cycling drivers. Carbon use efficiency is widely recognized as critical regulator of SOM and microbe dynamics[54,55] and is integrated in many microbial-explicit models. But C-STABILITY predicts that carbon use efficiency variation differently affects plant and necromass residues. This theoretical output requires experimental investigations to verify its accuracy, because of potential significant implications on soil properties related to SOM nature. In addition, the sensitivity analyses contradicted the general belief that enzyme traits have a limited effect on SOM chemistry and quantity[6,9]. It calls for closer consideration of the impact of catabolic processes on SOM cycling.

## Discussion

C-STABILITY combines compartmental and continuous modeling approaches. This enables the representation of SOM as a continuum of forms and the key processes governing its cycling, such as substrate accessibility and selective depolymerization, with a parsimonious number of parameters, which no models to date have been able to achieve. C-STABILITY breaks the abstract notion of quality of previous continuous models of SOM dynamics, which is closely associated with the recalcitrance concept and inadequate to account for several decomposers, and replaces it by an operational description of the organic matter chemistry in terms of biochemistry, polymerization, and physicochemical inaccessibility. This major change allows us to recognize that enzymes' and microbes' access to substrates strongly regulates SOM turnover. Degradation is indeed not solely determined by any intrinsic molecular recalcitrance or specific decay rate as in many models. It is determined by the spatial arrangement of soil components at very fine scale, the substrate biochemistry and polymerization, and the functional diversity of microbes producing enzymes. Jointly, they regulate substrate accessibility to enzyme and its selective depolymerization prior uptake.

The two modeling philosophies inspiring C-STABILITY are usually opposed and have their own limitations for long term predictions. Weakness of the continuous approach is typical of the statistical modeling. Despite the strong predictive ability of statistical models for stable systems, the lack of explicit description of functional processes lowers the confidence in predictions for perturbed systems. In opposite, compartment approach is descriptive and process based. But, the more the system is complex, the more process-based approach is facing an inflation of its parameters number and uncertainty, making long-term

prediction hazardous. C-STABILITY is intended to avoid these pitfalls by representing some of the key processes of the substrate-microbe system and by depicting the depolymerization statistically, as a continuous process. A detailed description of SOM depolymerization with compartmental model would require as many pools as classes of polymer sizes chosen by the modeler and many more parameters to describe the fluxes between all the pools of polymer size classes. Instead, the continuous distribution reports a wave of carbon traveling through the depolymerization process (Fig. 2). The moment a polymer becomes accessible to enzymes, the carbon it contained begins to flow through less and less polymerized forms. This journey is parsimoniously described by the $\alpha_{enz}$ parameter, which represents the substrate cleavage type (see[56] for a similar approach). In addition, to minimize the number of parameters, C-STABILITY does not account for the immense variety of enzymes involved in SOM decomposition, nor for the very high diversity of microbial populations. Enzymes are grouped in families and microbes in functional communities, both according to their action on SOM dynamics. Consequently, only a few parameters are necessary to characterize a wide range of enzyme and microbe characteristics. This parsimony is crucial for the clarity and robustness of the model.

At this stage of the model development it is essential to verify, through simple cases, that the fundamental assumptions of the model are relevant and allow an accurate description of observed processes. For this reason we set the model parameters at a constant value ($e_{mic}^0$, $u_{mic}^0$...) while we could assign a dependence on substrate polymerization properties or environmental conditions such as moisture or temperature. There is strong experimental evidence that carbon use efficiency is dependent on the substrate biochemistry[55,57]. Nevertheless, through simple cases, we demonstrated that the C-STABILITY framework successfully represents the current understanding of SOM cycling processes. In addition, by assembling recent mechanistic knowledge gained in the fields of biogeochemistry and microbial ecology, C-STABILITY generates novel insights on SOM cycling. It highlights the sensitive role of the depolymerization processes on SOM dynamics. It generates the temporal changes in substrate polymerization distribution as a function of enzyme characteristics, and elucidates how these changes impact decomposer communities succession. It revealed how substrate decay or persistence, potentially in an altered form, depends on positive or competitive interactions between the actors of decomposition (enzymes and microbes). A goal of C-STABILITY is also to produce predictions of C stocks and dynamics over the long term in scenarios with change in plant inputs quality and in microbial functional diversity, e.g., land use change or crop/tree species conversion. This could greatly support the design of demanding and expensive long-term experiments. The way forward is now to validate the model on datasets obtained for all the C pools considered in C-STABILITY (total-C, $CO_2$, microbial-C, biochemical pools), first in controlled conditions. Further development of the model should then address the impact of environmental variables on microbe ecology and biogeochemical processes and report substrate association with minerals. C-STABILITY framework appears particularly adapted to report the building and disruption of organo-mineral assemblages because substrate chemistry strongly influences such processes[33]. The frequency of the transfer from an inaccessible mineral-associated pool to an accessible one, or vice versa, would be dependent on substrate chemistry and also on a few soil properties such as moisture, pH, and clay content, known to influence organo-mineral interactions[58].

The model formalism has the advantage of offering extended possibilities for subsequent development stages. It is open to the introduction of competition relationships between decomposers and additional limiting factors such as water and nutrient availability. Besides, the equations describing C fluxes, such as

microbe mortality and enzyme catalytic rate, were chosen here to minimize the number of parameters. Yet other formulations could also be adopted. For example, the current mortality rate formulation, which assumes linear dependence on the microbial biomass as in many microbial models, exhibits a decadal oscillatory behavior in response to C input perturbations, while demonstrating little or no sensitivity of steady-state C to changes in C input[59]. To avoid this behavior, C-STABILITY could implement a density-dependent formulation of mortality[60,61]. The enzyme catalytic rate could also be modulated by a saturating function such as the reverse Michaelis–Menten equation[61], even though this was not required here to ensure model stability as in earlier models[62].

Finally, the operational description of SOM chemistry and decomposer catabolic activity in C-STABILITY framework represents an asset to strengthen our mechanistic knowledge on SOM dynamics. It offers a platform to identify knowledge locking points, to shape interactions between soil scientists and microbiologists, and to stimulate discussions on the mechanistic controls of SOM cycling. Interdisciplinary collaborations could support the development of soil-specific ecosystem management strategies to successfully enhance soil C sequestration and climate mitigation.

## Methods
### C-STABILITY description
*Organic matter representation.* The description of SOM in C-STABILITY consists of several subdivisions. First, organic matter is separated in two main pools, one for living microbes (noted $C_{mic}$) and one for the substrate (noted $C_{sub}$) (Fig. 1a). Several groups of living microbes can be considered simultaneously (e.g., bacteria, fungi, etc.) and C-STABILITY classes them into functional communities. Second, SOM is also separated between several biochemical classes, e.g., cellulose (or plant sugar), lignin, lipid, protein, and microbial sugar in this study (Fig. 1a). Third for each biochemical class, substrate accessible to its enzymes (noted ac) is separated from substrate which is inaccessible (noted in) due to specific physicochemical conditions, e.g., interaction between different molecules, inclusion in aggregates, sorption on mineral surfaces, etc.

Polymerization is a driver of interactions between substrate and living microbes and a continuous description of the degree of organic matter polymerization (noted $p$) is provided for each of these pools, as a distribution (Fig. 1b). The polymerization axis is oriented from the lowest to the highest degree of polymerization. A right-sided distribution corresponds to a highly polymerized substrate whereas a left-sided distribution corresponds to monomer or small oligomer forms. For each biochemical class $*$ ($* =$ cellulose, lignin, lipid, etc.), the polymerization range is identical for both accessible and inaccessible pools. The total amount of C (in $g_C$) in the accessible and the inaccessible pools of any biochemical class is as follows:

$$C_*^{ac} = \int_{p_*^{min}}^{p_*^{max}} \chi_*^{ac}(p)dp, \tag{1}$$

$$C_*^{in} = \int_{p_*^{min}}^{p_*^{max}} \chi_*^{in}(p)dp, \tag{2}$$

where $p_*^{min}$ and $p_*^{max}$ are the minimum and maximum degrees of polymerization of the biochemical class $*$ and $\chi_*^{ac}$, $\chi_*^{in}$ ($g_C \cdot p^{-1}$) are the polymerization distributions. Finally, the total substrate C pool is defined as the sum of all biochemical pools,

$$C_{sub} = \sum_* (C_*^{in} + C_*^{ac}). \tag{3}$$

Accessibility to microbe uptake is described by the interval (also called domain) $\mathcal{D}_u$, which corresponds to small substrate compounds, monomers, dimers or trimers smaller than 600 Daltons[1], that microbes are able to take up (in red in Fig. 1b). Besides, accessibility to enzymes occurs in the $\mathcal{D}_{enz}$ domain (in blue in Fig. 1b). Over time the substrate accessible to enzymes is depolymerized and its distribution shifts toward $\mathcal{D}_u$ where it eventually becomes accessible to microbe uptake.

The numerical rules chosen to represent polymerization are as simple as possible in the context of theoretical simulations. Each pool is associated with a polymerization interval $[p_*^{min}, p_*^{max}]$ of length two. Initial substrate distributions are represented by Gaussian distributions centered at a relative distance of 25% from $p_*^{max}$ (here 0.5), with the standard deviation set at 5% of polymerization interval length (here 0.1). In the accessible pool, the microbial uptake domains $\mathcal{D}_u$ are positioned at the left of the interval with a relative length of 20% (here 0.4), and enzymatic domains $\mathcal{D}_{enz}$ overlap the entire polymerization intervals.

*Organic matter dynamics.* As described in Fig. 1, three processes drive OM dynamics: (i) enzymatic activity, (ii) microbial uptake, biotransformation, and

mortality, and (iii) changes in local physicochemical conditions. First, enzymes have a depolymerization role, which enables the transformation of highly polymerized substrate into fragments accessible to microbes. Second, microbial uptake of substrate is only possible for molecules having a very small degree of polymerization. When C is taken up, a fraction is respired and the remaining is metabolized, and biotransformed into microbial molecules that return to the substrate upon microbe death. Each microbial group has a specific signature that describes its composition in terms of biochemistry and polymerization. Third, changes in local substrate conditions drive exchanges between substrate accessible and inaccessible to enzymes (e.g., aggregate formation and break). All of these processes are considered with a daily time step (noted $d$).

**Enzymatic activity** Enzymes are specific to biochemical classes. They are not individually reported, but rather as a family of enzymes contributing to the depolymerization of a biochemical substrate (e.g., combined action of endoglucanase, exoglucanase, betaglucosidase, etc., on cellulose will be reported as cellulolytic action). Figure 2 describes how substrate polymerization distributions are impacted by enzymes. The overall functioning of each enzyme family (noted enz) is described by two parameters: a depolymerization rate $\tau_{enz}^0$ providing the number of broken bonds per time unit and a factor accounting for the type of substrate cleavage $\alpha_{enz}$. The term $F_{enz}^{act}$ ($g_C \cdot p^{-1} \cdot d^{-1}$) represents the change in polymerization of $\chi_*^{ac}$ due to enzyme activity for all $p \in \mathcal{D}_{enz}$,

$$F_{enz}^{act}(\chi_*^{ac}, p, t) = -\tau_{enz}(t)\chi_*^{ac}(p, t)$$
$$+ \int_{\mathcal{D}_{enz}} \mathcal{K}_{enz}(p, p')\tau_{enz}(t)\chi_*^{ac}(p', t)dp'. \tag{4}$$

The depolymerization rate, $\tau_{enz}$ ($d^{-1}$), is expressed as a linear function of microbial C biomass $C_{mic}$ ($g_C$),

$$\tau_{enz}(t) = \tau_{enz}^0 C_{mic}(t), \tag{5}$$

where $\tau_{enz}^0$ ($g_C^{-1} \cdot d^{-1}$) is the action rate of a given enzyme per amount of microbial C. If several microbial communities are associated with the same enzyme family, we replace the $C_{mic}$ term by a weighted sum of the C mass of all communities involved in Eq. (5). The $\mathcal{K}_{enz}$ ($p^{-1}$) kernel provides the polymerization change from $p'$ to $p$,

$$\mathcal{K}_{enz}(p, p') = \mathbb{1}_{p \leq p'}(\alpha_{enz} + 1)\frac{(p - p_*^{min})^{\alpha_{enz}}}{(p' - p_*^{min})^{\alpha_{enz}+1}}, \tag{6}$$

where $\mathbb{1}_{p \leq p'}$ equals 1 if $p \leq p'$ and 0 otherwise. The $\alpha_{enz}$ cleavage factor denotes the enzyme efficiency to generate a large amount of small fragments and to shift the substrate polymerization distribution toward the microbe uptake domain $\mathcal{D}_u$ (Fig. 2). $\alpha_{enz} = 1$ is typical of the action of endo-cleaving enzymes, which randomly disrupts any bond of its polymeric substrate and generates oligomers. The shift toward $\mathcal{D}_u$ is slower if $\alpha_{enz}$ increases. This is characteristic of exo-cleaving enzymes, which attack the end-members of their polymeric substrate, generate small fragments, and preserve highly polymerized compounds. To satisfy the mass balance, the kernel verifies $\int \mathcal{K}_{enz}(p, p')dp = 1$. Then $F_{enz}^{act}$ does not change the total C mass but only the polymerization distribution (i.e., $\int F_{enz}^{act}(\chi, p, t)dp = 0$).

**Microbial biotransformation** Each microbial group (denoted mic) produces new organic compounds from the assimilated C. After death, the composition of the necromass returning to each biochemical pool $*$ of SOM is assumed to be constant, accessible and is depicted with a set of distributions $s_{mic,*}$, named signature. Each distribution $s_{mic,*}$ ($p^{-1}$) describes the polymerization of the dead microbial compounds returning to the pool $*$. The signature is normalized and unitless to ensure mass conservation, i.e., if we note that,

$$S_{mic,*} = \int_{p_*^{min}}^{p_*^{max}} s_{mic,*}(p)dp, \tag{7}$$

then we have $\Sigma_* S_{mic,*} = 1$.

For each accessible pool of substrate, the term $F_{mic,*}^{upt}$ describes how the microbes utilize the substrate available in the microbial uptake $\mathcal{D}_u$ domain (Fig. 1b). For all $p \in \mathcal{D}_u$,

$$F_{mic,*}^{upt}(\chi_*^{ac}, p, t) = u_{mic,*}^0 C_{mic}(t)\chi_*^{ac}(p, t), \tag{8}$$

where $u_{mic,*}^0$ ($g_C^{-1} \cdot d^{-1}$) is the uptake rate per amount of microbe C. The substrate uptake rate linearly depends on the microbial C quantity.

Depending on a carbon use efficiency parameter $e_{mic,*}^0$ (ratio between microbe assimilated C and taken up C), taken up C is respired or assimilated and biotransformed into microbial metabolites. This induces a change in the biochemistry and polymerization (Fig. 1a).

Finally, microbial necromass returns to the substrate pools with a specific mortality, which linearly depends on the microbial C quantity,

$$F_{mic,*}^{nec}(p, t) = m_{mic}^0 C_{mic}(t)s_{mic,*}(p), \tag{9}$$

where $m_{mic}^0$ ($d^{-1}$) is the mortality rate of the microbe (Fig. 1a).

**Change in local physicochemical conditions** The polymerization of a substrate inaccessible to its enzymes remains unchanged over time. A specific event changing the accessibility to enzymes (e.g., aggregate disruption or desorption from mineral surfaces) is modeled with a flux from the inaccessible to the accessible pool.

Transfer between these pools is described by the $F_{\mathrm{ac},*}^{loc}$ term for each biochemistry $*$,

$$F_{\mathrm{ac},*}^{loc}(p,t) = \tau_{\mathrm{tr}}^{\mathrm{ac}} \chi_*^{\mathrm{in}}(p,t) \qquad (10)$$

where $\tau_{\mathrm{tr},*}^{\mathrm{ac}}$ $(\mathrm{d}^{-1})$ is rate of local condition change toward accessibility.

Transfer in the opposite way (e.g., aggregate formation, association with mineral surfaces) is described by the $F_{\mathrm{in},*}^{loc}$ term,

$$F_{\mathrm{in},*}^{loc}(p,t) = \tau_{\mathrm{tr}}^{\mathrm{in}} \chi_*^{\mathrm{ac}}(p,t) \qquad (11)$$

where $\tau_{\mathrm{tr},*}^{\mathrm{in}}$ $(\mathrm{d}^{-1})$ is rate of local condition change toward inaccessibility.

**Organic matter input** We defined time dependent distributions for carbon input fluxes. There are denoted $i_*^{\mathrm{ac}}$ and $i_*^{\mathrm{in}}$ $(\mathrm{g_C.p^{-1}.d^{-1}})$ for both accessible and inaccessible pools of biochemical classes $*$. The total carbon input flux, expressed in $\mathrm{g_C.d^{-1}}$, is:

$$I(t) = \sum_* \int_{p_*^{\min}}^{p_*^{\max}} \left( i_*^{\mathrm{in}}(p,t) + i_*^{\mathrm{ac}}(p,t) \right) dp. \qquad (12)$$

**General dynamics equations** The distribution dynamics for each biochemical class $*$ is obtained from Eqs. (4)–(6) and (8)–(11),

$$\frac{\partial \chi_*^{\mathrm{ac}}}{\partial t}(p,t) = F_{\mathrm{ac},*}^{loc}(p,t) - F_{\mathrm{in},*}^{loc}(p,t)$$
$$+ F_{\mathrm{enz}}^{\mathrm{act}}(\chi_*^{\mathrm{ac}},p,t) + \sum_{\mathrm{mic}} \left( F_{\mathrm{mic},*}^{\mathrm{nec}}(p,t) - F_{\mathrm{mic},*}^{\mathrm{upt}}(\chi^{\mathrm{ac}},p,t) \right) + i_*^{\mathrm{ac}}(p,t), \qquad (13)$$

$$\frac{\partial \chi_*^{\mathrm{in}}}{\partial t}(p,t) = F_{\mathrm{in},*}^{loc}(p,t) - F_{\mathrm{ac},*}^{loc}(p,t) + i_*^{\mathrm{in}}(p,t). \qquad (14)$$

Then, the expended equations are,

$$\frac{\partial \chi_*^{\mathrm{ac}}}{\partial t}(p,t) = \tau_{\mathrm{tr},*}^{\mathrm{ac}} \chi_*^{\mathrm{in}}(p,t) - \tau_{\mathrm{tr},*}^{\mathrm{in}} \chi_*^{\mathrm{ac}}(p,t)$$
$$- \tau_{\mathrm{enz}}^0 \sum_{\mathrm{mic}} C_{\mathrm{mic}}(t) \chi_*^{\mathrm{ac}}(p,t)$$
$$+ \tau_{\mathrm{enz}}^0 \sum_{\mathrm{mic}} C_{\mathrm{mic}}(t)(\alpha_{\mathrm{enz}}+1)$$
$$\int_p^{p_*^{\max}} \frac{(p - p_*^{\min})^{\alpha_{\mathrm{enz}}}}{(p' - p_*^{\min})^{\alpha_{\mathrm{enz}}+1}} \chi_*^{\mathrm{ac}}(p',t) dp' \qquad (15)$$
$$+ \sum_{\mathrm{mic}} C_{\mathrm{mic}}(t) m_{\mathrm{mic}}^0 s_{\mathrm{mic},*}(p)$$
$$- \sum_{\mathrm{mic}} C_{\mathrm{mic}}(t) \mathbb{1}_{\mathcal{D}_u}(p) u_{\mathrm{mic},*}^0 \chi_*^{\mathrm{ac}}(p,t)$$
$$+ i_*^{\mathrm{ac}}(p,t),$$

$$\frac{\partial \chi_*^{\mathrm{in}}}{\partial t}(p,t) = \tau_{\mathrm{tr},*}^{\mathrm{in}} \chi_*^{\mathrm{ac}}(p,t) - \tau_{\mathrm{tr},*}^{\mathrm{ac}} \chi_*^{\mathrm{in}}(p,t) + i_*^{\mathrm{in}}(p,t). \qquad (16)$$

where $\mathbb{1}_{\mathcal{D}_u}(p)$ equals 1 if $p \in \mathcal{D}_u$ and 0 otherwise.

The dynamics of the total substrate is ruled by,

$$\frac{dC_{\mathrm{sub}}(t)}{dt} = \sum_* \int_{p^{\min}}^{p^{\max}} \left( \frac{\partial \chi_*^{\mathrm{ac}}}{\partial t}(p,t) + \frac{\partial \chi_*^{\mathrm{in}}}{\partial t}(p,t) \right) dp. \qquad (17)$$

The dynamics of microbial $C_{\mathrm{mic}}$ is obtained by,

$$\frac{dC_{\mathrm{mic}}}{dt}(t) = -m_{\mathrm{mic}}^0 C_{\mathrm{mic}}(t) + \sum_* u_{\mathrm{mic},*}^0 e_{\mathrm{mic},*}^0 C_{\mathrm{mic}}(t) \int_{\mathcal{D}_u} \chi_*^{\mathrm{ac}}(p,t) dp, \qquad (18)$$

and the $CO_2$ flux $(\mathrm{g_C.d^{-1}})$ produced by the microbes is given by,

$$F_{CO_2}(t) = C_{\mathrm{mic}}(t) \sum_* u_{\mathrm{mic},*}^0 \left( 1 - e_{\mathrm{mic},*}^0 \right) \int_{\mathcal{D}_u} \chi_*^{\mathrm{ac}}(p,t) dp. \qquad (19)$$

*Model implementation.* The model was implemented in the Julia© language[63,64]. An explicit finite difference scheme approximates the solutions of integro-differential equations with a $\Delta t = 0.1d$ time step and a $\Delta p = 0.01p$ polymerization step. Differential equations were solved with a Runge–Kutta method.

## Scenarios

*Scenario 1: cellulose decomposition kinetics and model sensitivity.* A first simulation was run to depict cellulose depolymerization and uptake by a decomposer community over one year (see parameters in Table 1). A global sensitivity analysis focusing on the residual cellulose variable was made to determine (i) the relative influence of parameters, and (ii) how parameters influence varies over time (Fig. 3c). A specific attention was given on enzymatic parameters (especially $\alpha$) to verify the pertinence of their introduction in the model.

We considered a specific method defined by Sobol for calculating sensitivity indices[65]. It provides the relative contribution of the model parameters to the total model variance, here at different times of the simulation. The method relies on the same principle as the analysis of variance. It was designed to decompose the variance of a model output according to the various degrees of interaction between the $n$ uncertain parameters $(x_i)_{i \in \{1,n\}}$. Formally, by assuming that the parameter

uncertainties are independent, the model output, denoted $y$, could be expressed as a sum of functions that take parameter interactions into account,

$$y = f_0 + \sum_{i=1}^n f_i(x_i) + \sum_{\substack{i,j=1 \\ i \neq j}}^n f_{i,j}(x_i,x_j) + ... + f_{1,...,n}(x_1,...,x_n). \qquad (20)$$

Under independence assumptions between models parameters variability, model variance is:

$$\mathbb{V}ar(y) = \sum_{i=1}^n \mathbb{V}ar(f_i(x_i))$$
$$+ \sum_{\substack{i,j=1 \\ i \neq j}}^n \mathbb{V}ar(f_{i,j}(x_i,x_j)) \qquad (21)$$
$$+ ... + \mathbb{V}ar(f_{1,...,n}(x_1,...,x_n)).$$

This variance decomposition leads to the definition of several sensitivity indices. The first-order Sobol's index of each parameter is,

$$S_i = \frac{\mathbb{V}ar(f_i(x_i))}{\mathbb{V}ar(y)}, \qquad (22)$$

and higher order indices are defined by:

$$S_{i,j} = \frac{\mathbb{V}ar(f_{i,j}(x_i,x_j))}{\mathbb{V}ar(y)}, \qquad (23)$$

and so on. These indices are unique, with a value of 0–1 and their sum equals 1. Here we focused on Sobol's first-order indices as they are usually sufficient to give a straightforward interpretation of the actual influence of different parameters[66,67]. We computed the sensitivity of the model outputs at several times of the simulation to highlight the role of model's parameters at different phases. Figure 3c shows the normalized Sobol's first-order indices to illustrate the relative influence of the model parameters on residual cellulose-C amount. Sobol's indices were estimated using a Monte Carlo estimator of the variance[68]. This was performed for a small variation in parameter values (±5% uniform variability), by running 12,000 model simulations for the Monte Carlo sampling.

*Scenario 2: effect of substrate inaccessibility to enzyme on litter decomposition kinetics.* A simulation of lignocellulose (76% cellulose, 24% lignin) degradation was performed by taking into account peroxidases, which deconstruct the lignin polymer, and cellulases, which hydrolyze cellulose. The cellulose was initially embedded in lignin and inaccessible to cellulase. The lignolytic activity (peroxidases) induces a disentanglement of the cellulose from the lignocellulosic complex. Therefore, the action of peroxidases was seen as a change of cellulose physicochemical local conditions resulting in a progressive transfer to the accessible pool. This transfer was assumed to be linearly related to the activity of lignolytic enzymes in Eq. (10),

$$\tau_{\mathrm{tr,cell.}}^{\mathrm{ac}} = \tau_{\mathrm{tr,cell.}}^{\mathrm{ac},0} \int_{\mathcal{D}_{\mathrm{lig.}}} \tau_{\mathrm{lig.}}^0 C_{\mathrm{mic}}(t) \chi_{\mathrm{lig.}}^{\mathrm{ac}}(p,t) dp, \qquad (24)$$

where $\mathcal{D}_{\mathrm{lig.}}$ is the domain of lignolytic activity and where the $\tau_{\mathrm{tr,cell.}}^{\mathrm{ac},0}$ coefficient is set at 13 $\mathrm{g_C^{-1}}$ for the current illustration.

The simulation was performed over one year. Enzymatic and microbial parameters given in Table 1 were chosen to be closely in line with the litter decomposition and enzyme action observation[16,37,69].

*Scenario 3: effect of community succession on C fluxes and substrate biochemistry.* We simulated the succession of two microbial functional communities, on the same previous lignocellulose, considering microbial residue recycling. The parameters (Table 1) were chosen according to the microbial community succession observations[43,45,70]. The first microbial community was specialized in plant substrate degradation, the second was specialized in the degradation of microbial residues. We referred to them as plant decomposers and microbial residue decomposers. Microbial residue decomposers were more competitive than plant decomposers because of their higher carbon use efficiency and lower mortality rate (Table 1). Both communities had the same biochemical signature, i.e., 50% polysaccharides, 30% lipids, and 20% proteins. We tested the impact of cheating as follows. Either uptake was impossible, i.e., $u_0$ equaled 0 for the community not involved in enzyme production, or uptake was possible but at a lower rate than the enzyme producers because the substrate fragments were released in the vicinity of the enzyme producers (Table 1).

*Scenario 4: soil organic matter composition at steady state.* We resolved the analytic formulation of the C stock and chemistry at steady state under several assumptions. We only considered one microbial community, a continuous constant plant input $I$ at a rate of $2.74.10^{-4} \mathrm{g_C.cm^{-2}.d^{-1}}$ and microbial recycling[47]. To be able to explicitly calculate the steady state, we only considered accessible pools, then cellulose substrate was not embedded in lignin but directly accessible to cellulolytic enzymes (Fig. 6). Finally, we considered that C use efficiency ($e_{\mathrm{mic}}^0$) and uptake

$(u_{mic}^0)$ parameters were identical for all biochemical classes. A full mathematical proof is given in Supplementary Note 3.

At steady state, the amount of microbial carbon is,

$$C_{mic} = \frac{e_{mic}^0 I}{(1 - e_{mic}^0) m_{mic}^0}. \tag{25}$$

For each biochemical class $*$, we define $p_*^u$ which verifies $p_*^{min} < p_*^u < p_*^{max}$ and $\mathcal{D}_u = [p_*^{min}, p_*^u]$, and

$$\theta_*(p) = m_{mic}^0 s_*(p) + \frac{i_*(p)}{C_{mic}}. \tag{26}$$

By considering the scenario setup (Table 1) and Eq. (26), if $* =$ plant sugar and lignin, $\theta_*(p) = m_{mic}^0 \frac{(1 - e_{mic}^0) i_*(p)}{e_{mic}^0 I}$ and if $* =$ lipid, protein and microbial sugar, $\theta_*(p) = m_{mic}^0 s_*(p)$. The steady-state distribution of C is obtained for $p \in [p_*^u, p_*^{max}]$,

$$\chi_*^{ac}(p) = \frac{\theta_*(p)}{\tau_{enz}^0} + \frac{\alpha_{enz} + 1}{p - p_*^{min}} \int_p^{p_*^{max}} \frac{\theta_*(p')}{\tau_{enz}^0} dp', \tag{27}$$

and for $p \in [p_*^{min}, p_*^u[$,

$$\chi_*^{ac}(p) = \frac{\theta_*(p)}{\tau_{enz}^0 + u_{mic}^0} + (\alpha_{enz} + 1) \frac{(p_*^u - p_*^{min})^{\beta - 1}}{(p - p_*^{min})^\beta} \int_{p_*^u}^{p_*^{max}} \frac{\theta_*(p')}{\tau_{enz}^0 + u_{mic}^0} dp'$$
$$+ (\alpha_{enz} + 1) \frac{\tau_{enz}^0}{\tau_{enz}^0 + u_{mic}^0} \int_p^{p_*^u} \frac{(p' - p_*^{min})^{\beta - 1}}{(p - p_*^{min})^\beta} \frac{\theta_*(p')}{\tau_{enz}^0 + u_{mic}^0} dp', \tag{28}$$

where $\beta = \frac{\tau_{enz}^0 - \alpha_{enz} u_{mic}^0}{\tau_{enz}^0 + u_{mic}^0}$.

Compared to previous scenarios related to laboratory experiments, enzymatic activities were divided by five to account for the impact of in situ climatic conditions. Other enzymatic and microbial parameters are given in Table 1. To explore how the catabolic traits of enzymes and anabolic traits of microbes affect the SOM composition, we modified individual parameters by 50% and documented their effects on the steady-state amount of C and its biochemistry.

## Data availability

Data sharing not applicable to this article as no experimental datasets were generated during the current study.

## Code availability

All code is available from the corresponding author upon request and C-STABILITY code is available at https://github.com/juliensaintemarie/C-STABILITY[63].

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

## Acknowledgements
We gratefully acknowledge financial support from the Laboratory of Excellence ARBRE (ANR-11-LABX-0002-01). We also thanks Matthias Cuntz and Jérôme Balesdent (deceased and to whom we pay tribute) for their advice.

## Author contributions
S.-M.J., B.M., S.-A.L., and D.D. designed the model. S.-M.J. and B.M. formulated the model equations. S.-M.J. implemented the numerical code. S.-M.J., B.M., and D.D. analyzed the model outputs with feedback from S.-A.L., G.E., and M.F. to ensure the ecological relevance of the scenarios. S.-M.J. and D.D. wrote the present article with critical inputs from B.M., S.-A.L., G.E., and M.F. All authors gave final approval for publication.

## Competing interests
The authors declare no competing interests.
