## [Peer Review File · Nature Communications]

REVIEWER COMMENTS

Reviewer #1 (Remarks to the Author):

This is an interesting and thought-provoking extension of older models of soil carbon. It goes beyond standard pool models by challenging their descriptions as homogenous and continuous models by adding microbial dynamics explicitly. Yet, it comes at a price, more parameters are required and the task of testing it against observations of soil C dynamics that rarely include more than total C is not obvious.

Specific comments

1. Line 119. It is strange that microbial uptake rate had no effect. It should change the growth rate of microbes and hence microbial biomass. Explain. Is it a consequence of other parameter choices that make the quantities of C available for uptake very small?
2. Figure 6a. It is not clear what the meaning of the vertical dashed line are. I suppose the solid lines represent the distributions over degree of polymerization.

Reviewer #2 (Remarks to the Author):

This study by Julien et al. detailed the process of substrate degradation by exoenzymes via representing depolymerization as a continuous process, as opposed to the prevailing scheme of either Michaelis-Menten or reverse MM-based on degradation. The presentation is in such a high quality, but I do not quite agree with the message this piece is trying to articulate.

Although mechanistically this model pushes forward one process in the SOM-Microbes system, substrate polymerization, toward being more explicit by incorporating the insights of theoretical and experimental understanding of SOM's continuous nature, I do not believe this work with this model elucidates any process with new insights in the carbon-microbes system.

The study highlighted substrate accessibility in influencing carbon turnover, which, however, is a recognition existing for not a short time period. So I am hesitated to say this study contributed much to our understanding with enough novelty. In addition, from the perspective of modelling accuracy in soil systems carbon dynamics, this work lacks model-data comparison. Therefore, it's hard or still early to claim this model is better than others. From both the perspective of shedding new light on processes and the perspective of improving simulation accuracy, this work did not convince me with enough novelty.

In detail, this model makes substrate degradation explicit by treating it as a continuous process with an introduction of parameters including cleavage, and min and max of depolymerization of each substrate. On the one hand, these parameters introduced more uncertainty. At the same time, representing the microbial community with coarse guilds and microbial cell metabolism with a parameter CUE (which should be an emergent property) is mechanistically a step backwards relative to existing models that already make community relatively more explicit. This tradeoff in development makes me reluctant to accept that making one process explicit while sacrificing the explicitness of other processes can warrant a claim of a better model developed, at least for now without a systematic comparison. For example, how do we know models using MM or reverse MM to capture substrate degradation is worse in capture system dynamics than this substrate-continuum one?

Reviewer #3 (Remarks to the Author):

General comments:

Julien and co-authors present an interesting modeling paper that documents a novel approach for representing enzyme and microbial decomposition of organic matter substrates. My chief concern, however, is that at its core this is really a model development paper that may be more appropriate for a more discipline specific journal that will allow more space for an in-depth description (and review) of the model's structure, assumptions and parameterizations (e.g. SBB, FEMS, or ISME?).

My second concern is that with a structure and parameterization as complicated as C-STABILITY should be able to be configured to capture many of the behaviors that are illustrated in the text. These capabilities ARE interesting, but it also makes me think the model is likely over-parameterized for application at larger scales. Specifically, it's not clear from me how one moves beyond these nice idealized experiments to actually simulate this complexity of across multiple sites for long periods of time (which seems to be implied in the abstract and discussion)? This can be rectified by clearly managing reader (and reviewer) expectations from the start. The model captures a bunch of really interesting behavior related to enzyme depolymerization to microbial community dynamics in theoretical space. It also helps clarify some key uncertainties or and assumptions in the model that could be validated with future experimentation. In my mind these are the strengths of the paper and revisions are warranted to help make these insights clearer.

Specific and technical comments:

In my estimation the title borrows too heavily from the Lehmann and Kleber (2015) paper and should be more original.

Lines 7-8 this sentence is phrased awkwardly and can be edited for clarity.

Lines 10-11 I'm not sure where this feature of the model is demonstrated in the manuscript, and while I understand this is the aim of the authors it has not yet been shown and suggest removing the sentence.

Lines 49-52. While it's true that compartment models cannot describe a continuum of decay or enzyme diversity, it has also not been shown that this level of detail is actually necessary to capture litter decay or SOM stabilization dynamics. I would avoid making this kind of logical fallacies when justifying the need for the approach taken here. This comes up again in lines 84-85. And while it's nice to be able to simulate the "organic forms generated by enzyme depolymerization", I'm not sure this is critical to improving our projections on SOM dynamics under climate change?

Fig 1. I'm sure there are some nuances, but at first glance this model structure looks very similar to the CORPSE model (Sulman et al 2014), with 5 OM pools here (instead of 3 in CORPSE).

Line 73, what are 'lowly polymerized molecules'?

Line 74-75 details of how the model handles the spatial arrangement of substrates and minerals in the soil matrix seems important to describe here, especially if the aim is to make projections at larger spatial and temporal scales?

Fig 1b, 2a, 3a. I don't understand the units of the y-axis (is this gC/pool)?

Line 68-85, The introduction includes too much description of the details of the model in my estimation.

Line 86 & section 4.3 I don't think "scenarii" is a word.

Line 86, the key questions should be clarified and appropriate publications cited.

Section 2 Results: For what seems like a model documentation paper it seems crazy to jump into results. I'm not sure the format of this journal is well suited for the aims of the paper.

For each of the results (e.g. Line 104 & Fig 3, line 138 & Fig 4; line 201 & Fig 6) it seems showing observations would be valuable there too if trying to validate the model (as implied in the text).

Details of how results in Fig. 3c were generated would be helpful. These findings seems interesting, but the methods are too sparse to understand or evaluate.

Line 134 is embedment a word?

Fig 4b, I'm not really clear on the theory here, but why would microbes quickly degrade inaccessible cellulose, but not touch the accessible cellulose in this simulation?

Line 204, it seems the details of how the C-STABILITY handles mineral association are critical for the long-term projections from the model, but missing from the manuscript.

Fig 6b. There's a wealth of information crammed into this figure that is sparingly described and barely interpreted. What are readers supposed to take home from this display item?

Line 231, I'm not sure what "a parsimonious number of parameters" is intended to convey, but I worry that the model may be over-parameterized for broad-scale application. It's also not clear to me why it's critical to simulated this level of detail related to 'substrate accessibility and selective depolymerization".

Line 241-247, While I agree with this assessment of continuous and compartment model classes, it's not clear to me how C-Stability avoids the pitfalls of either approach (or indeed inherits them both)!

Lines 282-286. I'm intrigued about the specifics of how this could be done. The text jumps from global-scale aspirations to a discussion of proteomics and metabar coding and then back to Earth system prediction. I wonder how a model like C-STABILITY helps to bridge that mismatch in scales?

References:

Sulman, B. N., Phillips, R. P., Oishi, A. C., Shevliakova, E., & Pacala, S. W. (2014). Microbe-driven turnover offsets mineral-mediated storage of soil carbon under elevated CO₂. *Nature Climate Change*, 4(12), 1099-1102. doi:10.1038/nclimate2436

Reviewer #4 (Remarks to the Author):

Review for Sainte-Marie et al.

Sainte-Marie and co-authors report on the development of a novel model to describe organic matter decomposition dynamics. Their work differs from previous models in that organic matter pools are described as polymerization length distributions which are modified by enzymes that can exhibit distinct preference for endo- and exo-cleavage. The authors studied the models behavior in four scenarios spanning a gradient of complexity from single-substrate decomposition to a complete organic soil layer.

The authors address an important topic of great interest to the research community – how can organic matter decomposition be described mathematically. Their approach has important advantages compared to previous models and holds great potential for improving our understanding of this important process. The manuscript is well written and clearly of great interest to the readership.

The authors' model is at an early development stage, and the authors focus on demonstrating the overall validity of the model and its potential in simple scenarios. This means that many processes and relations that are common to soils are not incorporated into the model (e.g., effects/limitations of nutrient availability of enzyme production and microbial substrate use efficiency, potential changes in the abundance of exo- and endo-cleaving enzymes at different decomposition stages, difference in the accessibility of organic matter to exo- and endo-cleaving enzymes). However, the authors make it clear in the discussion that they aim for a low level of complexity at this stage of the model development.

I think that the manuscript is in a great shape and only minor revisions are necessary for publication.

In my opinion, there are two ways in which the manuscript could be further improved:

- the manuscript describes the behavior of the model, but rarely discusses what the model tells about the modeled system. The manuscript could thus be improved by more explicitly discussing how the model results presented within this manuscript contribute novel insights to our understanding of decomposing organic matter/soil systems.

- generally, the manuscript text could be edited to further ease readability. While some complexity in the writing is unavoidable when describing mathematical formulations, I think carefully editing the text for during a revision for easier reading would improve the reach of the manuscript. In particular, it feels like many of the figures are only scarcely explained/referred to in the main text, and it's not always clear why a certain dataset is depicted.

Minor comments:

L191: some comment is needed why the impossible C content of 1.5 gC/gsoil is a reasonable result. I would suggest that C content on a weight or volume content basis is a poor way to describe organic soil layers, where the total weight/volume of the soil (e.g., per m² surface) changes depending on the amount of organic material present (i.e., gC/gsoil in these systems remains fairly constant around 0.5, with differences in OM amounts changing the organic layer thickness rather than it's carbon content.)

L326: "Each microbial group" instead of "each microbe"

Lukas Kohl

Responses to the reviewers

Reviewer #1 (Remarks to the Author):

This is an interesting and thought-provoking extension of older models of soil carbon. It goes beyond standard pool models by challenging their descriptions as homogenous and continuous models by adding microbial dynamics explicitly. Yet, it comes at a price, more parameters are required and the task of testing it against observations of soil C dynamics that rarely include more than total C is not obvious.

The Authors: Thank you for your comment. We are confident that the model testing and further broader utilization will benefit from the rising number of experimental studies, which jointly monitor different C pools (total-C, CO₂, microbial-C, biochemical pools).

Reviewer #1: 1. Line 119. It is strange that microbial uptake rate had no effect. It should change the growth rate of microbes and hence microbial biomass. Explain. Is it a consequence of other parameter choices that make the quantities of C available for uptake very small?

The Authors: The referee is right, microbial uptake rate has no effect because of the other parameters values. We clarified this in the Result section adding a sentence on line 119-121.

Reviewer #1: 2. Figure 6a. It is not clear what the meaning of the vertical dashed line are. I suppose the solid lines represent the distributions over degree of polymerization.

The Authors: The vertical dashed lines delimit the microbial uptake domain for each biochemical substrate. We clarified this point in the caption of figure 6.

Reviewer #2 (Remarks to the Author):

This study by Julien et al. detailed the process of substrate degradation by exoenzymes via representing depolymerization as a continuous process, as opposed to the prevailing scheme of either Michaelis-Menten or reverse MM-based on degradation. The presentation is in such a high quality, but I do not quite agree with the message this piece is trying to articulate.

Although mechanistically this model pushes forward one process in the SOM-Microbes system, substrate polymerization, toward being more explicit by incorporating the insights of theoretical and experimental understanding of SOM's continuous nature, I do not believe this work with this model elucidates any process with new insights in the carbon-microbes system.

The study highlighted substrate accessibility in influencing carbon turnover, which, however, is a recognition existing for not a short time period. So I am hesitated to say this study contributed much to our understanding with enough novelty. In addition, from the perspective of modelling accuracy in soil systems carbon dynamics, this work lacks model-data comparison. Therefore, it's hard or still early to claim this model is better than others. From both the perspective of shedding new light on processes and the perspective of improving simulation accuracy, this work did not convince me with enough novelty.

In detail, this model makes substrate degradation explicit by treating it as a continuous process with an introduction of parameters including cleavage, and min and max of depolymerization of each substrate. On the one hand, these parameters introduced more uncertainty. At the same time, representing the microbial community with coarse guilds and microbial cell metabolism with a parameter CUE (which should be an emergent property) is mechanistically a step backwards relative to existing models that already make community relatively more explicit. This tradeoff in development makes me reluctant to accept that making one process explicit while sacrificing the explicitness of other processes can warrant a claim of a better model developed, at least for now without a systematic comparison. For example, how do we know models using MM or reverse MM to capture substrate degradation is worse in capture system dynamics than this substrate-continuum one?

The Authors: We thank the referee for raising the important point about the novelty of the present model, which was not fully addressed in the initial version. Below we also provide clarifications about the model development stage and the associated trade-offs. Then, we take into account the referee's concerns regarding substrate accessibility, microbial guilds and Michaelis-Menten's formulation.

About the novelty of the present model.

As stated in the first lines of our rebuttal letter, **we now underline how C-STABILITY can be used to infer novel mechanistic insights on SOM cycling**, while its use to improve large scale projection through our model outputs has been toned down. Indeed, some original predictions emerge from our assemblage of the recent mechanistic knowledge gained in different disciplines.

For this purpose, at the end of the introduction (lines 82-92) we list in the revision some currently pending research questions about the decomposition mechanisms, for which C-STABILITY theoretical framework/predictions could be used to tackle/modelize these questions.

In the Results section, a short paragraph provides for each scenario the novel outputs generated by our model predictions.

- a. scenario 1: C-STABILITY captures the known heterogeneity of organic matter forms and generates a distribution of the degree of polymerization for different biochemical types of SOM, which may be demanding to obtain experimentally and is not produced by any model yet. The model also demonstrates that catabolic processes are sensitive regulators of SOM degradation, while they are often overlooked.
- b. Scenario 2 illustrates how the coordinated action of secreted enzymes governs the kinetics of complex substrate decay. This scenario provides the opportunity to guide the design of future experiments to test novel research questions related to the depolymerization. (For example, related to the degradative strategies developed by rotting fungi – white rot vs. brown rot).
- c. Scenario 3 reveals how the succession of microbial decomposer communities is impacted by the chemistry of the decaying substrate. It additionally reveals that a substrate may persist in soil in a partly depolymerized form due to decomposers competition, a result that would be hardly accessible by classical methods of wet chemistry.
- d. Scenario 4 highlights the overlooked role of catabolic processes in SOM dynamics (as scenario 1) and provides original results on the distinct impact of CUE on plant and microbial residues persistence.

In the discussion we briefly summarize these outputs on lines 300-307.

We also mention on lines 307-311 that a key feature of C-STABILITY is its ability to produce long term predictions for scenarios strongly sensitive to the novel mechanisms framed in its structure (i.e. depolymerization, functionality of decomposer communities). The inferences emerging from theoretical scenarios analyses will be particularly helpful when molecular and biochemical experiments should be implemented over several months (or even years), sometimes in situ or/and with continuous monitoring of CO₂ emission, resulting in high costs for experiment maintenance.

Thus, we expect that the innovative C-STABILITY framework will facilitate the sharing of recent knowledge between different disciplines (for example, the importance of microbial communities ecology and interaction may be an underestimated aspect for some geochemists, while the substrate inaccessibility to enzyme related to physico-chemical substrate embedment in mineral aggregate or organic matrix may be under-estimated by some microbiologists) **and stimulate discussions on the mechanistic controls of SOM cycling.** (see concluding section lines 322-328)

Finally, as identified by the referee, the novelty also lies in the modeling approach, which challenges the pool models in their homogeneous description and the continuous models by explicitly representing microbial dynamics.

The referee regretted that we did neither present any fit to real data nor compare the model predictions to another model predictions. Indeed, what is proposed is the first step in the development of a novel modelling approach. It consists in moving forwards from the concepts and model assumptions into mathematical formulations, production of theoretical scenarios and selection of parameters according to the diverse co-authors expertises. This first step should ultimately demonstrate that the model accurately reproduces C cycling processes and dynamics as they are currently understood. The next step in our model development will be its confrontation to real data jointly obtained for all the C pools considered in C-STABILITY, (total, CO₂, microbial-C, biochemical pools). (comment added line 312)

At this stage, to demonstrate the accuracy of model prediction, we did not try to match precisely a given experiment, (notably because some experiments illustrate the variability of decomposition pattern or SOM composition at steady state e.g. Kaffenberger and Schilling 2015; Balaria and Johnson 2013) but is inspired by the general pattern of several studies. To clarify our approach, we now present a Table in Supplementary Material that summarizes the references we used to parameterize the model and how we utilized them.

Such a model development inevitably requires **trade-offs**. The referee underlines that we moved a step backwards with regard to some recent models that make microbial communities description and functional quite explicit. But as resumed by Wieder et al. GCB 2018, the priority to improve confidence in SOM cycling prediction requires balancing demands between formulating model structures that adequately represent the current understanding of processes and avoiding too complex models. The balance is obtained by making choices about the assumptions on the processes and factors regulating SOM cycling.

The C-STABILITY model focuses on decomposer action on SOM depolymerization and biotransformation, which represents to our knowledge an important step forwards in the representation of current understanding of SOM processes and contributes to the novelty of the work. This necessarily implies to degrade the description of some other important processes/factors, which become implicit, to avoid complexity and the subsequent increase in model uncertainty. We do recognize the significance of models making microbial community physiology and functioning explicit. Our goal here is not to claim that our model makes a better representation of substrate degradation than other models. It offers another framework, more adapted for example to support the resolution of research issues about the biochemical quality and level polymerization of SOM and how they may impact C sequestration.

From a more general point of view, the diversity of model structures should be viewed as a chance to improve confidence in model predictions (i.e. the IPCC community jointly run a dozen of models to assess the future C stocks - see also Shi et al. Nature Com. 2018).

Specific points:

About accessibility, it has indeed long been recognized as one of the pathways of soil protection. What is novel in C-Stability is the distinction between substrate accessibility to enzymes and substrate accessibility to microbe uptake (see introduction on lines 71-72). We also change the title of 2.1 to underline the notion of accessibility to uptake in contrast to the accessibility to enzyme in title 2.2.

About the suggestion of the referee to include a MM or reverse MM function in the uptake term.

This is an option that we already considered, see comment on line 329-331 in the discussion, and we may opt for it in the future as it is technically easily feasible. It will be made at the price of a couple of additional parameters but will interestingly smooth/attenuate the slope of substrate loss on Figure 4 and introduce a sensitivity of steady state stocks to input quantity.

Specifically, about guild. We were too restrictive by mentioning guilds. This term has been removed and replaced by microbe functional communities. Indeed, the model can be implemented at any

level description of microbial communities, why not at the level of individual species. Whatever the level of details in the decomposer description, the model requires for each decomposer: the substrate(s) altered by this decomposer, its depolymerization pattern and catalytic activity summarized by the alpha and tau parameters, the biochemical signature of the necromass and three fitness parameters (uptake rate, mortality rate and CUE).

references:

Wieder, W. R., Hartman, M. D., Sulman, B. N., Wang, Y. P., Koven, C. D., & Bonan, G. B. (2018). Carbon cycle confidence and uncertainty: Exploring variation among soil biogeochemical models. *Global change biology*, 24(4), 1563-1579.

Shi, Z., Crowell, S., Luo, Y., & Moore, B. (2018). Model structures amplify uncertainty in predicted soil carbon responses to climate change. *Nature communications*, 9(1), 1-11.

Reviewer #3 (Remarks to the Author):

General comments:

Julien and co-authors present an interesting modeling paper that documents a novel approach for representing enzyme and microbial decomposition of organic matter substrates.

My chief concern, however, is that at its core this is really a model development paper that may be more appropriate for a more discipline specific journal that will allow more space for an in-depth description (and review) of the model's structure, assumptions and parameterizations (e.g. SBB, FEMS, or ISME?).

My second concern is that with a structure and parameterization as complicated as C-STABILITY should be able to be configured to capture many of the behaviors that are illustrated in the text. These capabilities ARE interesting, but it also makes me think the model is likely over-parameterized for application at larger scales. Specifically, it's not clear from me how one moves beyond these nice idealized experiments to actually simulate this complexity across multiple sites for long periods of time (which seems to be implied in the abstract and discussion)? This can be rectified by clearly managing reader (and reviewer) expectations from the start.

The model captures a bunch of really interesting behavior related to enzyme depolymerization to microbial community dynamics in theoretical space. It also helps clarify some key uncertainties or and assumptions in the model that could be validated with future experimentation. In my mind these are the strengths of the paper and revisions are warranted to help make these insights clearer.

The Authors: To reinforce the model description, we dissociated the mathematical presentation of the model and the details of methods used for each scenario. For instance, the sensitivity analysis description of the methods is now included in the scenario 1 part. We also added some equations in the Methods section to clarify some steps of the model building process (i.e. Equations 2, 3, 7, 12, 13, 14 and 17). We moved the table of parameters from the supplementary information into the material and methods section. In Supplementary Material we now provide the full proof of steady state equation calculation.

As suggested, we also clarified in the revision the main goals of our modeling work. We removed the mention to large scale predictions. We rather emphasized on the new insights allowed by the C-STABILITY model in the theoretical space and how they may stimulate/support the design of future experimentations (this was also a request of referees #2 and #4). For this purpose, we added a few sentences at the end of each scenario analysis in the Result section. We also briefly recall in the discussion that the assemblage in C-STABILITY of the novel paradigms in microbial ecology and biogeochemistry generate novel insights on SOM cycling and pave the way forwards for novel investigations. (see details in the response to referee 2, section about model novelty)

Specific and technical comments:

In my estimation the title borrows too heavily from the Lehmann and Kleber (2015) paper and should be more original.

The Authors: The reference to the title by Lehmann and Kleber (2015) was intentional, but we may suggest the following alternatives

C-STABILITY: an innovative modeling framework to leverage the continuous representation of organic matter.

or,

C-STABILITY: an innovative modeling framework to portray the continuum of soil organic matter forms and dynamics.

Lines 7-8 this sentence is phrased awkwardly and can be edited for clarity.

The Authors: This sentence has been removed as we reworked the abstract according to the referees' suggestions on the work novelty and outcomes.

Lines 10-11 I'm not sure where this feature of the model is demonstrated in the manuscript, and while I understand this is the aim of the authors it has not yet been shown and suggest removing the sentence.

The Authors: We removed the sentence. Instead we exposed that the model may support in the future the exploration of novel mechanistic hypotheses and the design of experiments to investigate them (lines 11-12).

Lines 49-52. While it's true that compartment models cannot describe a continuum of decay or enzyme diversity, it has also not been shown that this level of detail is actually necessary to capture litter decay or SOM stabilization dynamics. I would avoid making this kind of logical fallacies when justifying the need for the approach taken here. This comes up again in lines 84-85. And while it's nice to be able to simulate the "organic forms generated by enzyme depolymerization", I'm not sure this is critical to improving our projections on SOM dynamics under climate change?

The Authors: We removed the two sentences mentioned by Referee #3. And as already explained, we clarified the strengths of the model. Instead of driving the reader's expectations towards improved SOM dynamics predictions, we highlight the mechanistic insights generated by the model and how the model can be utilized to support the discussion of novel mechanistic hypotheses and the design of future experiments.

Fig 1. I'm sure there are some nuances, but at first glance this model structure looks very similar to the CORPSE model (Sulman et al 2014), with 5 OM pools here (instead of 3 in CORPSE).

The Authors: CORPSE and C-STABILITY indeed both emphasize on substrate inaccessibility and chemistry. We now mention this similarity on line 155
Nevertheless C-STABILITY identifies precise biochemical classes and reports the degree of substrate polymerization, what offers a mechanistic frame to report functional diversity of microbe and the catalytic action of enzymes

Line 73, what are 'lowly polymerized molecules'?

The Authors: We changed this term to “small oligomers” (line 72). These oligomers could be oligosaccharides or small peptides resulting from the degradation of cellulose (and other polysaccharides) or proteins, respectively.

Lines 74-75 details of how the model handles the spatial arrangement of substrates and minerals in the soil matrix seems important to describe here, especially if the aim is to make projections at larger spatial and temporal scales?

The Authors: At this stage of model development, we focus indeed our attention on purely organic systems. As underlined by the referee, the dynamics of mineral organic associations will be key processes to reproduce to perform long term prediction, and this is one of the next close perspectives for C-STABILITY further development. The C-STABILITY framework appears particularly tailored for such an ambition because it is well established that organo-mineral interactions are driven by substrate chemistry. We underline this in the text (lines 314-320).

Fig 1b, 2a, 3a. I don't understand the units of the y-axis (is this gC/pool)?

The Authors: The unit “*p*” stands for polymerization. We make it explicit in every figure caption to avoid any misunderstanding.

Line 68-85, The introduction includes too much description of the details of the model in my estimation.

The Authors: In the initially submitted version, we tried to provide the minimum of details in the Introduction section but still wanted to present the major features of the model. This choice was motivated by the necessity for the readers to understand which mechanistic assumptions are embedded in the C-STABILITY framework and what is its novelty compared to other models. In our view, the description provided in the first version correctly balances between too many and not enough pieces of information. We still removed two sentences (previously on lines 71-72 and 77-79) and are open to any suggestion to further streamline this introduction.

Line 86 & section 4.3 I don't think “scenarii” is a word.

The Authors: “Scenarios” has indeed a more common usage than “scenarii”. We performed the change.

Line 86, the key questions should be clarified and appropriate publications cited.

The Authors: We modified the text to list the key questions addressed in our 4 scenarios (lines 82-86) and included a Supplementary Table listing the publications used to make the various scenarios realistic.

Section 2 Results: For what seems like a model documentation paper it seems crazy to jump into results. I'm not sure the format of this journal is well suited for the aims of the paper.

The Authors: We now clarified from the beginning that the aim of the paper is not only the novel modeling approach but also the new insights provided by model theoretical simulations and how the approach can be used to move forwards in the understanding of SOM processes.

For each of the results (e.g. Line 104 & Fig 3, line 138 & Fig 4; line 201 & Fig 6) it seems showing observations would be valuable there too if trying to validate the model (as implied in the text).

The Authors: We clarify in the revision that our goal is not to validate the model, but to make the simulations realistic and in line with the general patterns found in publications. This is now mentioned on line 87.

This choice is notably supported by some of the references we used to support the scenarios production, listed in the new Supplementary Table (such as Kaffenberger et al., 2015 or Balaria 2013), which demonstrate the diversity of decomposition kinetics or of SOM composition at steady state and because validation of the simulations would require C data for all the pools considered in C-STABILITY (biochemical pools, microbe biomass, CO₂).

Details of how results in Fig. 3c were generated would be helpful. These findings seems interesting, but the methods are too sparse to understand or evaluate.

The Authors: Methods and Figure caption were modified to give a better explanation of the sensitivity analysis of Figure 3c.

Line 134 is embedment a word?

The Authors: Embedment is an English word, which means the act of embedding or the state of being embedded.

According to the English proofreader we solicited, “embedment” would be a better choice than “embedding” in the sentence where it is used. Nevertheless, if the referee considers we should definitely opt for embedding, we will follow his/her advice.

Fig 4b, I’m not really clear on the theory here, but why would microbes quickly degrade inaccessible cellulose, but not touch the accessible cellulose in this simulation?

The Authors: Obviously our figure was not clear enough. Microbes are not degrading inaccessible cellulose. The amount of inaccessible cellulose-C quickly decreases because the lignin physical barrier is altered by lignolytic activity. This implies a quick transfer of C from the inaccessible cellulose pool to the accessible cellulose pool. An arrow was added in Figure 4b to show the possible exchanges between inaccessible and accessible pools of cellulose (as in Figure 1). The caption has also been modified to explain how the C transfer occurs between accessible and inaccessible cellulose pools.

Line 204, it seems the details of how the C-STABILITY handles mineral association are critical for the long-term projections from the model but missing from the manuscript.

The Authors: As already mentioned above, the simulation of mineral organic associations dynamics will indeed be critical processes to reproduce to perform long term prediction. We keep the focus of the novel version on purely organic systems as C-STABILITY already provides a lot of theoretical results on such systems.

However, we provided a few sentences about how C-STABILITY would handle organo-mineral associations on lines 315-420 and indicate on line 314 that it is one of the future developments of our model.

Fig 6b. There's a wealth of information crammed into this figure that is sparingly described and barely interpreted. What are readers supposed to take home from this display item?

The Authors: We modified Figure 6 to improve its readability. For some reasons we made a mistake in uploading this figure in the first submitted version, and it was not consistent with the text. We deeply apologize for that. In the revision, the text has been amended to ease the figure understanding and interpretation (lines 230-234 and 237-239). The Figure caption has also been improved.

Line 231, I'm not sure what "a parsimonious number of parameters" is intended to convey, but I worry that the model may be over-parameterized for broad-scale application. It's also not clear to me why it's critical to simulated this level of detail related to 'substrate accessibility and selective depolymerization".

The Authors: The notion of parsimony is relative to how a process-based approach would describe the changes in polymer size due to depolymerization. In C-STABILITY four parameters (α , τ , p_{min} and p_{max}) are needed to describe the whole process of depolymerization for a given biochemical pool. Describing the same phenomenon with a compartment model will require as many pools as classes of polymer sizes chosen by the modeler and many more parameters to describe the fluxes between all the pools of polymer size classes. We clarified it on lines 278-279 and 282-284.

C-STABILITY would indeed be over-parameterized for large scale application and it would be inappropriate to go so deep into mechanisms for large scale predictions. Following the reviewer's recommendation, we modified the text by removing any reference to the potential use of C-STABILITY to improve large scale prediction. Nevertheless, we provide below some indications about how we consider this could be done (see the response to the referee's last comment).

Line 241-247, While I agree with this assessment of continuous and compartment model classes, it's not clear to me how C-Stability avoids the pitfalls of either approach (or indeed inherits them both)!

The Authors: The major pitfall of the compartmental model approach is the inflation of parameters when going deeper into the description of mechanisms. C-STABILITY is not affected by this parameter inflation because it is based on a continuous representation to describe depolymerization (what is parsimonious in terms of parameters as previously explained) and we grouped microbes in functional communities and enzymes into families (see modified sentence line 289).

Besides, C-STABILITY intends to avoid the pitfall of statistical models, which are limited to predict the behavior of systems in transition, because it explicitly describes some key-processes of the substrate-microbe system. (see lines 278-282).

Lines 282-286. I'm intrigued about the specifics of how this could be done. The text jumps from global-scale aspirations to a discussion of proteomics and metabarcoding and then back to Earth system prediction. I wonder how a model like C-STABILITY helps to bridge that mismatch in scales?

The Authors: As mentioned above, we deleted in the revision any mention to global scale aspirations for the model. As suggested by the reviewer, we rather focused on the novel insights provided by model theoretical simulations and how the approach can be used to move forwards in the understanding of SOM processes. However, to answer your specific comment here, we think that C-STABILITY could be useful to improve large scale models by identifying emergent microbial/enzyme drivers of SOM dynamics. C-STABILITY could generate region and soil type-specific patterns of decomposition the substrate nature. They could then be injected in broad scale models. Nevertheless, these thoughts are very preliminary. They have to mature and be tested. For these reasons they have been removed from the paper.

Reference: Sulman, B. N., Phillips, R. P., Oishi, A. C., Shevliakova, E., & Pacala, S. W. (2014). Microbe-driven turnover offsets mineral-mediated storage of soil carbon under elevated CO₂. *Nature Climate Change*, 4(12), 1099-1102. doi:10.1038/nclimate2436

Reviewer #4 (Remarks to the Author):

Sainte-Marie and co-authors report on the development of a novel model to describe organic matter decomposition dynamics. Their work differs from previous models in that organic matter pools are described as polymerization length distributions which are modified by enzymes that can exhibit distinct preference for endo- and exo-cleavage. The authors studied the models behavior in four scenarios spanning a gradient of complexity from single-substrate decomposition to a complete organic soil layer.

The authors address an important topic of great interest to the research community – how can organic matter decomposition be described mathematically. Their approach has important advantages compared to previous models and holds great potential for improving our understanding of this important process. The manuscript is well written and clearly of great interest to the readership.

The authors' model is at an early development stage, and the authors focus on demonstrating the overall validity of the model and its potential in simple scenarios. This means that many processes and relations that are common to soils are not incorporated into the model (e.g., effects/limitations of nutrient availability of enzyme production and microbial substrate use efficiency, potential changes in the abundance of exo- and endo-cleaving enzymes at different decomposition stages, difference in the accessibility of organic matter to exo- and endo-cleaving enzymes). However, the authors make it clear in the discussion that they aim for a low level of complexity at this stage of the model development.

I think that the manuscript is in a great shape and only minor revisions are necessary for publication.

In my opinion, there are two ways in which the manuscript could be further improved:

- the manuscript describes the behavior of the model, but rarely discusses what the model tells about the modeled system. The manuscript could thus be improved by more explicitly discussing how the model results presented within this manuscript contribute novel insights to our understanding of decomposing organic matter/soil systems.

The Authors: We followed the referee's recommendation and clarified the model outputs, which was also suggested by referees #2 and #3. For this purpose, we included a few sentences at the end of each scenario analysis, highlighting how C-STABILITY simulations modify our understanding of SOM dynamics. In addition, we emphasize in the Discussion section that the assemblage of novel paradigms in microbial ecology and biogeochemistry in C-STABILITY generates novel insights on SOM cycling and paves the way forwards for novel investigations (see details in the response to referee #2, the section dealing with the model novelty).

- generally, the manuscript text could be edited to further ease readability. While some complexity in the writing is unavoidable when describing mathematical formulations, I think carefully editing the text during a revision for easier reading would improve the reach of the manuscript. In particular, it feels like many of the figures are only scarcely explained/referred to in the main text, and it's not always clear why a certain dataset is depicted.

The Authors: We improved the description of the Figures (e.g. on lines 182-183 for Figure 5 or lines 230-239 for Figure 6). We also clarified the Material and Methods section. We dissociated the mathematical presentation of the model and the details of methods used for each illustrative scenario. For instance, the sensitivity analysis description of the methods is now included in the scenario 1 part. We also added some equations to clarify some steps in the model building process.

Minor comments:

L191: some comment is needed why the impossible C content of 1.5 gC/gsoil is a reasonable result. I would suggest that C content on a weight or volume content basis is a poor way to describe organic soil layers, where the total weight/volume of the soil (e.g., per m² surface) changes depending on the amount of organic material present (i.e., gC/gsoil in these systems remains fairly constant around 0.5, with differences in OM amounts changing the organic layer thickness rather than its carbon content.)

The Authors: We thank the referee for pinpointing this mistake inherited from an earlier version. Currently units are g of C for stocks in scenarios 1, 2 and 3 and g of C per cm² for scenario 4. We carefully check that units were correct within the whole text and in the different figures.

L326: "Each microbial group" instead of "each microbe"

The Authors: Action done

Lukas Kohl

REVIEWERS' COMMENTS

Reviewer #2 (Remarks to the Author):

I appreciate the effort by the authors in providing such a detailed, well-constructed response to my concerns.

Among the many revisions, I particularly like the major changes made to the concerns regarding the novelty of this model. The authors further highlighted the novelty of building such a model to help elucidate the mechanisms underlying enzyme-organic matter interactions. Further, the authors toned down the claim of a "better" model in terms of capturing complicated soil carbon dynamics. Again, it COULD be, which, however, at this stage without model comparisons and data integration, cannot be judged. In short, underscoring its continuous modelling in potentially contributing to better understanding of carbon behavior may be a better selling point of this work, which, to me, is solid without overselling.

With these points considered, I do not have any more comments on this manuscript.

Reviewer #3 (Remarks to the Author):

I appreciate revisions made to this manuscript and largely only offer editorial suggestions to improve readability of the text.

I like the questions being used to guide the study (last paragraph of introduction) and assume that these questions correspond to each sections of the results (2.1-2.4). I would suggest that the: 1) Language used in result subheadings more closely correspond to the questions outlined in the introduction and 2) Authors explicitly address each question in the results. For example, from 2.1 make it painfully clear to readers "How catalytic processes a critical regulator of SOM decomposition" and "How the coordinated action of enzymes regulates complex substrate decomposition". NOTE, this is done in the Discussion (text ~ line 300), but I feel it would also be helpful in the results.

I also would take care to craft the questions guiding this paper carefully to illustrate the points you're really trying to make. For example, I would argue that we've known for a long time that catalytic processes are a critical regulator of SOM decomposition, indeed that's the one thing that the much-maligned discrete pool, microbial implicit models actually capture (see Schimel & Shaeffer 2012). Instead, it seems the new insight provided by C-STABILITY is what is the new insight you're trying to highlight the importance of depolymerization (and enzyme behavior) as a major rate limiting step controlling microbial uptake and decomposition of organic matter.

Line 62: Change to unrealistic

Line 80-81: It seems references needed to support this assertion

Line 83. Should this be enzyme or enzymes' (possessive)

Line 87, SI Table 1. Maybe also refer to the method here, as this is where 'scenarios' are all described.

SI Table 1, scenario 4 should be described as the "Chemical characterization of organic..."

Line 132, again should this be changed to enzymes' (possessive)?

Revisions to Fig 4 and section 2.2 are appreciated, but at first glance it still seems like the 'inaccessible' cellulose is 'decomposing' first (not just being transferred into the enzyme available

pool). This is more clear in the movie and I'm not sure what suggestions to make for Fig 4 to avoid confusion for readers that may be quickly looking through figures to get a sense for this work.

Line 148. What's the previous simulation (Fig 3)?

I'd recommend changing the last sentence of the Fig 5 caption. "When cheating occurs, plant decomposers are outcompeted by microbial residue decomposers, which results in the persistence of lignin.

Line 257-261. These conclusions seem to be an artifact of the configuration of the simulations. Without any mechanism for microbial residue persistence, it's not surprising that CUE and enzyme traits only effected the variation in plant residues turnover (which also form the bulk of steady-state SOM pools).

Line 270 should this be enzyme and microbial access to substrates, not the other way around, as currently written?

Line 271, I'm not sure I agree with this claim. In Table 1 τ_{enzyme} for cellulose > τ_{enzyme} for lignin. It seems like this parameterization results in the same net effect that cellulose has an intrinsically faster decomposition rate than lignin, as in most 'compartment models'.

Line 272. What are the "local environment properties" being referred to here? I would assume this includes abiotic factors like soil temperature, moisture availability, and mineralogy- none of which appear to modify the microbial activity or soil biochemistry from the simulations illustrated here. Maybe just leave out the text "local environment properties".

Line 298- as above, what are 'environmental conditions' here?

Line 308, Please replace 'key capacity' here with 'goal' or 'aim'. These long-term projections are an appropriate long-term goal for C-STABILITY, but I'm not sure this work demonstrates this capacity in the model.

Line 331: I'm pretty sure the Schimel and Weintraub model uses a reverse M-M equation (see also Buchkowski et al. 2017)

REFS: Buchkowski, R. et al (2017). Applying population and community ecology theory to advance understanding of belowground biogeochemistry. Ecology Letters, 20(2), 231-245.
doi:10.1111/ele.12712

Reviewer #4 (Remarks to the Author):

This is the revised version of a manuscript I have reviewed previously. I found that the manuscript was a very good state in the first place, and the authors have fully addressed all comments raised by me.

Response to reviewer 3

Reviewer 3 (Remarks to the Author):

I appreciate revisions made to this manuscript and largely only offer editorial suggestions to improve readability of the text.

I like the questions being used to guide the study (last paragraph of introduction) and assume that these questions correspond to each section of the results (2.1-2.4). I would suggest that the: 1) Language used in result subheadings more closely correspond to the questions outlined in the introduction and 2) Authors explicitly address each question in the results. For example, from 2.1 make it painfully clear to readers “How catalytic processes a critical regulator of SOM decomposition” and “How the coordinated action of enzymes regulates complex substrate decomposition”. NOTE, this is done in the Discussion (text ~ line 300), but I feel it would also be helpful in the results.

I also would take care to craft the questions guiding this paper carefully to illustrate the points you’re really trying to make. For example, I would argue that we’ve known for a long time that catalytic processes are a critical regulator of SOM decomposition, indeed that’s the one thing that the much-maligned discrete pool, microbial implicit models actually capture (see Schimel & Shaeffer 2012). Instead, it seems the new insight provided by C-STABILITY is what is the new insight you’re trying to highlight the importance of depolymerization (and enzyme behavior) as a major rate limiting step controlling microbial uptake and decomposition of organic matter.

The Authors: We thank the referee for his/her recommendation to clarify the points we are willing to demonstrate so as to better highlight the new insights brought by C-STABILITY .

- *Scenario 1. We modified the first question at the end of the introduction into “Is enzyme depolymerization a critical regulator of SOM decomposition?”
In the results subsection, we explicitly addressed the question and changed the sentences order in the last paragraph to highlight the new insight showing the importance of depolymerization on SOM decomposition. (Lines 97-100)*
- *Scenario 2. We modified the second question into “How is substrate accessibility to enzyme regulated?” and changed the results subheading into : “Coordinated action of enzymes regulates substrate accessibility”.
In the results, after reminding the question, we indicated that in the case of lignocellulose, this is the activity of one enzyme that provides a gateway to the substrate for another enzyme. (Lines 102-103) We added some words to highlight that the model provides quantitative pieces of information on the substrate accessibility regulation, which is more difficult to get experimentally. (Line 124)*
- *Scenario 3. We turned the question in the other way round: How does the succession of decomposer communities impact the chemistry of the decaying substrate?
(Instead of “How is the succession of decomposer communities impacted by the chemistry of the decaying substrate?”).
As reminded in the beginning of the result subsection, we addressed both questions in the third scenario, but we mainly focused on the decomposer impact on substrate chemistry, as also highlighted by the subheading. (Lines 132-133)*
- *Scenario 4. No change.*

Line 62: Change to unrealistic

The Authors: Change done

Line 80-81: It seems references needed to support this assertion:

The Authors: We slightly modified the sentence (line 59-60) and now refer to the Supplementary Table 1. This table provides references for degradation pathways by functional decomposer communities as they are currently understood - degradation pathways that are reproduced by C-STABILITY in the various presented simulations.

Line 83. Should this be enzyme or enzymes' (possessive)

The Authors: We thank the referee for this notification and modified the text.

Line 87, SI Table 1. Maybe also refer to the method here, as this is where 'scenarios' are all described.

The Authors: Change done

SI Table 1, scenario 4 should be described as the "Chemical characterization of organic..."

The Authors: We thank the referee for this notification. It was the French writing. Now this is corrected and written into English.

Line 132, again should this be changed to enzymes' (possessive)?

The Authors: Change done

Revisions to Fig 4 and section 2.2 are appreciated, but at first glance it still seems like the 'inaccessible' cellulose is 'decomposing' first (not just being transferred into the enzyme available pool). This is more clear in the movie and I'm not sure what suggestions to make for Fig 4 to avoid confusion for readers that may be quickly looking through figures to get a sense for this work.

The Authors: To improve the reader's understanding, we added "decomposable" or "non-decomposable" in the headings of the graphics showing the distribution in polymerization (subpanel a). We also added a reference to the Movie at the end of the Figure 4 caption.

Line 148. What's the previous simulation (Fig 3)?

The Authors: We added a reference to Figure 3.

I'd recommend changing the last sentence of the Fig 5 caption. "When cheating occurs, plant decomposers are outcompeted by microbial residue decomposers, which results in the persistence of lignin.

The Authors: We thank the referee for this notification, we modified the text accordingly.

Line 257-261. These conclusions seem to be an artifact of the configuration of the simulations. Without any mechanism for microbial residue persistence, it's not surprising that CUE and enzyme traits only affected the variation in plant residues turnover (which also form the bulk of steady-state SOM pools).

The Authors: We propose a novel version of the text to clarify the points about CUE and enzyme traits sensitivity addressed by the referee.

CUE sensitivity. (Lines 180-182) *The linear relationship between microbe biomass and decomposition activity implies that CUE affects both the turnover of plant and microbe compounds.*

Nevertheless CUE only affects the amount of plant residues at steady-state (as shown by the numerical sensitivity analysis and the steady-state equations 27, 28, S18 and S19). The modified turn-over of microbe residues is indeed compensated by modified microbe metabolite biosynthesis. In our view, this is not straightforward and may be of interest for the reader, even if selective preservation processes that are not currently implemented in C-STABILITY may further operate in mineral soil.

Enzyme traits sensitivity. (Line 190-196) *Enzyme traits affect plant residues but also microbe residues. We now provide examples for both instead of only giving values for cellulose.*

Line 270 should this be enzyme and microbial access to substrates, not the other way around, as currently written?

The Authors: We thank the referee for this notification and modified the text.

Line 271, I'm not sure I agree with this claim. In Table 1 τ_{enzyme} for cellulose > τ_{enzyme} for lignin. It seems like this parameterization results in the same net effect that cellulose has an intrinsically faster decomposition rate than lignin, as in most 'compartment models'.

The Authors: We agreed, we changed our sentence into "Degradation is indeed not solely determined by any intrinsic molecular recalcitrance or specific decay rate as in many models". (Line 211)

Line 272. What are the "local environment properties" being referred to here? I would assume this includes abiotic factors like soil temperature, moisture availability, and mineralogy- none of which appear to modify the microbial activity or soil biochemistry from the simulations illustrated here. Maybe just leave out the text "local environment properties".

The Authors: At this stage, we did not include temperature nor moisture in the model (even if they are important drivers of microbial activity. What we meant here is not the pedoclimate but the spatial arrangement of soil components at a very fine scale. It has been clarified in the text. (Lines 212- 213)

Line 298- as above, what are 'environmental conditions' here?,

The Authors: Here we meant pedoclimate, pH..., that we want to incorporate in the model in a close future. We clarified this point in the text. (Line 234)

Line 308, Please replace 'key capacity' here with 'goal' or 'aim'. These long-term projections are an appropriate long-term goal for C-STABILITY, but I'm not sure this work demonstrates this capacity in the model.

The Authors: We modified the text as suggested.

Line 331: I'm pretty sure the Schimel and Weintraub model uses a reverse M-M equation (see also Buchkowski et al. 2017)

REFS: Buchkowski, R. et al (2017). Applying population and community ecology theory to advance understanding of belowground biogeochemistry. Ecology Letters, 20(2), 231-245. doi:10.1111/ele.12712

The Authors: We thank the referee for this note. Schimel and Weintraub indeed used a reverse M-M equation. We corrected the text and also included a reference to Buchkowski et al. 2017.